# Spatial control of irreversible protein aggregation

Christoph Weber[1†], Thomas Michaels[1†], L Mahadevan[2,3,4]*

[1]School of Engineering and Applied Sciences, Harvard University, Cambridge, United States; [2]Department of Physics, Harvard University, Cambridge, United States; [3]Department of Organismic and Evolutionary Biology, Harvard University, Cambridge, United States; [4]Kavli Institute for NanoBio Science and Technology, Harvard University, Cambridge, United States

**Abstract** Liquid cellular compartments form in the cyto- or nucleoplasm and can regulate aberrant protein aggregation. Yet, the mechanisms by which these compartments affect protein aggregation remain unknown. Here, we combine kinetic theory of protein aggregation and liquid-liquid phase separation to study the spatial control of irreversible protein aggregation in the presence of liquid compartments. We find that even for weak interactions aggregates strongly partition into the liquid compartment. Aggregate partitioning is caused by a positive feedback mechanism of aggregate nucleation and growth driven by a flux maintaining the phase equilibrium between the compartment and its surrounding. Our model establishes a link between specific aggregating systems and the physical conditions maximizing aggregate partitioning into the compartment. The underlying mechanism of aggregate partitioning could be used to confine cytotoxic protein aggregates inside droplet-like compartments but may also represent a common mechanism to spatially control irreversible chemical reactions in general.
DOI: https://doi.org/10.7554/eLife.42315.001

*For correspondence:
lmahadev@g.harvard.edu

†These authors contributed equally to this work

Competing interests: The authors declare that no competing interests exist.

## Introduction

Spatial control within living cells is essential to many cellular activities, ranging from the local control of protein activity to the uptake of pathogens or the management of wastes (*Alberts, 2017*). Understanding the mechanisms underlying regulation of cell activities in space and time is key not only for biological function, but also in view of understanding and eventually controlling cellular dysfunction (*Knowles et al., 2011*; *Knowles et al., 2014*; *Chiti and Dobson, 2006*; *Gitler et al., 2017*; *Michaels et al., 2018*). The spatial organization of cellular activities is often associated with membrane-bound organelles that ensure permeation only for certain molecules of specific molecular structure (*Neupert and Herrmann, 2007*; *Wiedemann and Pfanner, 2017*; *Dukanovic and Rapaport, 2011*). Recently, new types of organelles have been discovered that do not possess a membrane. They are referred to as non-membrane-bound compartments and they share most hallmark properties with actual liquid-like droplets (*Brangwynne et al., 2009*; *Brangwynne, 2013*; *Elbaum-Garfinkle et al., 2015*; *Zhu and Brangwynne, 2015*; *Banani et al., 2017*). Unlike organelles surrounded by membranes, these non-membrane-bound compartments are formed by liquid-liquid phase separation. In many cases, this phase separation is driven by disfavoring interactions between the constituent molecules of the compartment and the surrounding cyto- or nucleoplasm (*Hyman et al., 2014*; *Brangwynne et al., 2015*). The partitioning of other intracellular molecules into such droplet-like compartments is then controlled by their relative interactions with the constituent molecules of the compartment.

These droplet-like compartments are ubiquitous inside living cells (*Banani et al., 2017*). For instance, they emerge prior to cell division (*Brangwynne et al., 2009*; *Parker and Sheth, 2007*),

and form as a response to cellular stress (*Patel et al., 2015*; *Malinovska et al., 2013*; *Molliex et al., 2015*). They have been shown to enrich proteins (*Hernández-Vega et al., 2017*; *Woodruff et al., 2017*; *Mateju et al., 2017*) and genetic material (*Parker and Sheth, 2007*; *Saha et al., 2016*; *Zhang et al., 2015*) providing distinct environments for chemical reactions and biological function. The molecules hosted inside these compartments may even be protected against other agents from the cytoplasm (*Franzmann et al., 2018*) or face conditions facilitating their molecular repair (*Mateju et al., 2017*; *Ganassi et al., 2016*; *Alberti et al., 2017*; *Alberti and Carra, 2018*; *Jain et al., 2016*; *Specht et al., 2011*). In addition to these roles, recent evidence suggests that liquid cellular compartments could play an important role in regulating pathological protein aggregation (*Alberti and Hyman, 2016*; *Shin and Brangwynne, 2017*). An example is the irreversible assembly of amyloids into fibrillar aggregates, a process that is linked to a large variety of currently incurable diseases (*Dobson, 2003*; *Knowles et al., 2014*; *Lashuel et al., 2002*; *Catalano et al., 2006*; *Benilova et al., 2012*; *Campioni et al., 2010*), such as Alzheimer's and Parkinson's diseases, amyloidosis or type-II diabetes. As another example, a chaperone in yeast uses a prion-like, intrinsically disordered domain to bind and sequester misfolded proteins in protein deposition sites (*Grousl et al., 2018*; *Boczek and Alberti, 2018*). Moreover, misfolded and pathological proteins can accumulate inside liquid-like stress granules triggering the aggregation kinetics inside these compartments. The presence of this phase separated compartment can promote the formation of fibrillar aggregates, and prevent aggregation outside the stress granules (*Molliex et al., 2015*; *Mateju et al., 2017*). Thus, the corresponding cytotoxic effects of protein aggregates are expected to be strongly localized in space as well. However, whether weak protein interactions are sufficient to significantly change the aggregate concentration in the compartment relative to homogeneous aggregation and how the physical parameters of aggregation and phase separation determine the partitioning of aggregates remains an open question.

Here, we combine the kinetics of irreversible protein aggregation with the theory of liquid-liquid phase separation to develop a model of irreversible assembly of protein fibrils in the presence of droplet-like compartments. We use this model to predict the partitioning of aggregates into the liquid compartment as a function of the fundamental physical parameters underlying aggregation kinetics and phase separation. We find that relatively weak interactions between the protein monomers and the liquid compartment molecules are sufficient to enrich the concentration of aggregates within the liquid compartment by several orders of magnitudes relative to homogeneous aggregation (*Figure 1*). This strong enrichment of aggregates emerges because the liquid compartment acts as continuous sink of monomers during the aggregation dynamics, thus promoting intra-compartment aggregation but suppressing aggregation outside of the compartment. Moreover, we find that aggregate partitioning is more pronounced for larger (smaller) compartments depending on the relative values of the reaction orders for primary and secondary nucleation. Our results suggest that cellular liquid compartments are ideal to control irreversible protein aggregation in space. In particular, the compartment volume, which is determined by the mean concentration of phase separated protein, represents a relevant control parameter for intra-compartment positioning of aggregate amount and size. The underlying physical mechanism might also be relevant in the context of spatial regulation of other irreversible chemical reactions where liquid compartments act as biomolecular microreactors.

## Model for liquid compartments controlling protein aggregation

To capture the interplay between liquid phase separation and protein aggregation kinetics we start with a model of two coexisting phases. One phase could be rich in proteins for example and coexist with a phase rich in another protein component, lipid, or water. Monomers that are prone to aggregate can partition differently into these phases. This partitioning is determined by the relative interactions between the majority components of each phase with the monomers. We consider the case where the partitioning of monomers is close to equilibrium during the kinetics of aggregation. This assumption is well justified since small, weakly interacting molecules such as the aggregating monomers diffuse between seconds and minutes through a cell of size in the order of tens of $\mu m$ (*Brangwynne et al., 2009*; *Griffin et al., 2011*), while typical time scales of aggregation in vitro are in the order of hours (see, for example, *Cohen et al., 2013*). Furthermore, the diffusion of aggregates is highly hindered as long fibrillar aggregates experience a much larger hydrodynamic drag force and can get entangled with cytoskeletal filaments and other assembled fibrils (*de Gennes,*

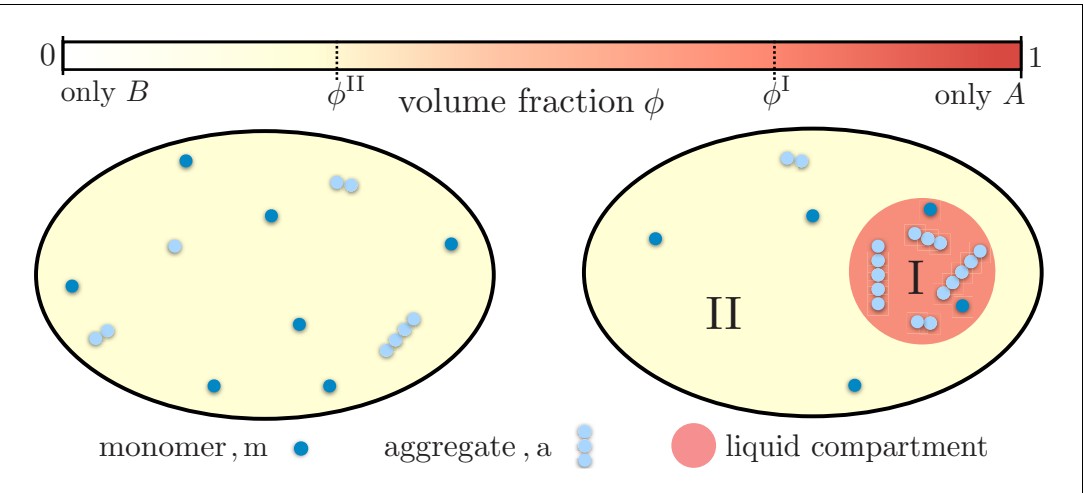

**Figure 1.** Partitioning of monomers and aggregates via liquid-like compartments. Protein aggregation may occur homogeneously inside cells also leading to aggregates inside more sensitive cellular regions (left). A liquid compartment may accumulate monomers and thereby trigger the local formation of aggregates (right). The hardly diffusing aggregates are thus kept away from a more sensible cellular region. Such a spatial segregation of aggregates is ideal for adding functional, drug-like molecules which dominantly dissolve inside the compartment. These molecules may degrade the aggregates or inhibit further growth and nucleation. But most importantly, as these molecules are localized inside the compartment their toxic effects are diminished.
DOI: https://doi.org/10.7554/eLife.42315.002

*1971*; *Rubinstein, 1987*). Finally, at large enough density and size, fibrils may even form solid-like gels (*Mateju et al., 2017*) further slowing down their mobility. All these effects imply that we may safely neglect diffusion of large aggregates and consider the typical case that monomers diffuse quickly relative to their aggregation kinetics.

We also consider the case where monomers and aggregates are dilute enough to neglect their influence on the composition of the two coexisting protein phases. Typical values of volume fractions for monomers of Amyloid-β, $c_\mathrm{m}\nu_\mathrm{m}$ (radius of gyration in the range 1–2 nm [*Sajfutdinow et al., 2018*]), at physiological concentrations between 100 pM to 1 nM are in the range of $10^{-9}$ to $10^{-8}$. Time scale separation and dilute monomers together ensure that the compartment can coexist at thermodynamic equilibrium while the partitioning kinetics of monomers may weakly deviate from the partitioning equilibrium. Thus, we first discuss the partitioning of monomers into phase separated compartments at equilibrium and then consider small deviations from this equilibrium to understand its consequences for protein aggregation.

## Phase separation and partitioning of monomers at equilibrium

We consider a system of total volume $V$ hosting a single liquid compartment (a droplet for example) of a condensed phase I of volume $V^\mathrm{I}$. The compartment itself forms by liquid-liquid phase separation between the two components $A$ and $B$. Compartment I is composed of the component $A$ and a small fraction of component $B$, while compartment II has a small amount of $A$ and a large amount of $B$, as depicted in *Figure 1*. Each compartment creates a distinct environment for the aggregating monomers.

For simplicity, we discuss the case of an incompressible system where the aggregating monomers 'm' and aggregates 'a' are dilute, that is $c_\mathrm{m}\nu_m \ll 1$ and $c_\mathrm{a}\nu_a \ll 1$, with $c_m$ and $c_a$ denoting the concentrations of monomers and aggregates and $\nu_m$ and $\nu_a$ are the respective molecular volumes. The assumption of dilute monomers and aggregates imply that for an incompressible system, the volume fractions $\phi_A$ and $\phi_B$ of the protein components $A$ and $B$ obey, $\phi_A + \phi_B = 1 - c_\mathrm{m}\nu_\mathrm{m} - c_a\nu_a \simeq 1$, where we abbreviate $\phi_A = \phi$ in the following. As a result, the monomers may partition differently into the respective minority and majority phases, but, due to their dilute concentrations, they do not affect the degree of phase separation. Under these circumstances and in the absence of binding processes, the partitioning of monomers in the two phases is governed by the relative interaction

strength $\Delta\chi$ between the monomers with the $A$ and the $B$ components, respectively. If $\Delta\chi$ is large and positive, monomers favor the presence of the majority component $A$ in compartment I. In this case, we expect a more pronounced partitioning of monomers into compartment I. Contrariwise, when $\Delta\chi$ is large and negative, monomers favorably partition into compartment II. The degree of monomer partitioning at equilibrium can be calculated using the condition that the chemical potentials of monomers associated with compartment I and II are balanced (see *Figure 2(a)*, and Appendix 1 for the derivation), and allows us to define the monomer partitioning

$$\Gamma \equiv \frac{c_m^I}{c_m^{II}} \simeq \exp\left[\frac{\nu_m}{\nu}\Delta\chi\left(\phi^I - \phi^{II}\right)\right], \tag{1}$$

where $c_m^I$, $c_m^I$ are the monomer concentrations in phases I and II, respectively, $\nu$ denotes the molecular volume of $A$ and $B$ molecules, and $\phi^I - \phi^{II} \in [0,1]$ is the degree of phase separation of the $A$-component. Then the relative partitioning of the total monomer concentration, $c_m^{tot} = \left(c_m^I V^I + c_m^{II} V^{II}\right)/V$, is given by the expressions $c_m^I = \xi(\bar{\phi})\Gamma c_m^{tot}$ and $c_m^{II} = \xi(\bar{\phi})c_m^{tot}$, where the partition degree

$$\xi(\bar{\phi}) = \frac{1}{1 + (\Gamma - 1)V^I(\bar{\phi})/V} \tag{2}$$

captures the impact of the relative size of the compartment volume $V^I(\bar{\phi})/V$. The volume of the compartment I is in turn controlled by the mean volume fraction $\bar{\phi}$ of $A$ molecules in the system in terms of the relationship $V^I(\bar{\phi}) = V(\bar{\phi} - \phi^{II})/(\phi^I - \phi^{II})$, where we neglected the volume contribution

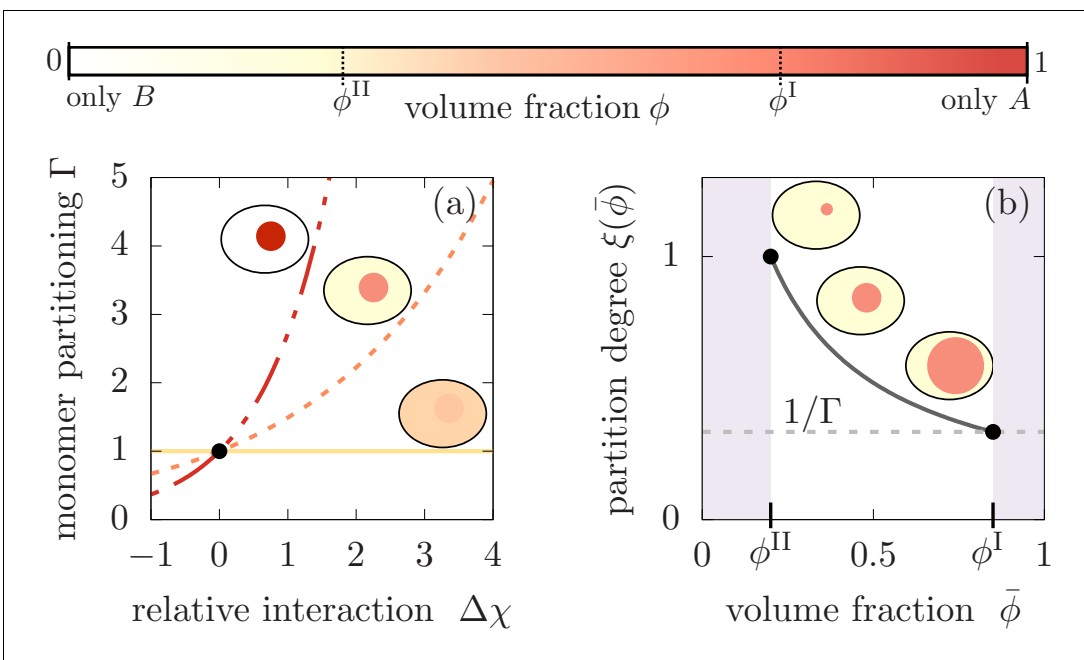

**Figure 2.** Monomer partitioning and relative degree of segregation. (**a**) The monomer partitioning $\Gamma$ (*Equation 1*) exponentially increases with the relative interaction strength $\Delta\chi$ (units of $k_BT$) between the monomers and the $A$ and $B$ molecules which is defined in the Appendix. Its characteristic increase is set by the degree of phase separation, $\phi^I - \phi^{II}$. Partitioning vanishes at the critical point of phase separation (solid line) and increases with the degree of phase separation (dashed line). Partitioning is largest for $\phi^I - \phi^{II} \simeq 1$ (dash-dotted line). Due to the exponential increase, large monomer partitioning $\Gamma$ can already be reached for weak relative interaction energies of a few $k_BT$. (**b**) The partition degree $\xi(\bar{\phi}) = c_m^{II}/c_m^{tot}$ (*Equation 2*) describing the concentration fraction of monomers that resides in the minority phase II of the compartment, decreases with the mean volume fraction of $A$ material, $\bar{\phi}$, along with increasing compartment volume $V^I(\bar{\phi})$. Smaller compartments are thus better in enriching the monomer mass concentration.
DOI: https://doi.org/10.7554/eLife.42315.003

of monomers and aggregates due to the considered dilute conditions. For finite sized compartments, the equilibrium volume fractions, $\phi^{\mathrm{I}}$ and $\phi^{\mathrm{II}}$, are slightly increased due to the Laplace pressure. However, for compartments significantly exceeding the size of the molecules the relative increase is weak and is thus neglected in the following (see Appendix 1).

## Model for protein aggregation coupled to non-equilibrium monomer partitioning

Due to the separation of time scales of monomer diffusion and monomer aggregation, the partitioning of monomers into the compartment is close to equilibrium at all times of the aggregation kinetics and thus the relative fraction of monomers is approximately governed by the monomer partitioning $\Gamma$, *Equation 1*. However, as the aggregation kinetics decreases the amount of monomers inside each phase, aggregation couples to the partitioning. This coupling is represented by a diffusive flux of monomer with a rate $J^\alpha$ in each phase, that attempts to maintain the monomer partitioning close to equilibrium. In the limit of a sharp interface separating the liquid compartment from the bulk, there is no aggregation at the interface, $J^{\mathrm{I}} = -J^{\mathrm{II}} \equiv J$. Furthermore, to linear order, the flux $J$ between the phases is proportional to the difference of monomer partitioning with respect the equilibrium value $\Gamma$ (see Appendix 2 for the derivation) and is of the form:

$$J = -k \left( M_{\mathrm{m}}^{\mathrm{I}} - \Gamma M_{\mathrm{m}}^{\mathrm{II}} \right), \tag{3a}$$

where $M_{\mathrm{m}}^\alpha = c_{\mathrm{m}}^\alpha m_{\mathrm{m}}$ (with $m_{\mathrm{m}}$ as monomer mass) is the monomer mass concentration in compartment $\alpha = \mathrm{I}, \mathrm{II}$, and $k$ denotes the rate at which monomer partitioning relaxes back to the equilibrium given by *Equation 1*. For simplicity, we consider the case where diffusion of monomers is constant and equal in each phase, and not affected by the aggregates.

Very generally, in a homogeneous solution, irreversible protein aggregation results from the combined action of several microscopic events, including (i) primary nucleation, whereby monomers spontaneously interact to form the smallest stable aggregate structures, (ii) fibril elongation, and (iii) secondary (i.e. aggregate-dependent) nucleation processes (*Michaels and Knowles, 2014*; *Michaels et al., 2016*; *Arosio et al., 2016*; *Michaels et al., 2018*; *Törnquist et al., 2018*). Secondary nucleation mechanisms (*Törnquist et al., 2018*) have been found to be active in many aggregating protein systems, ranging from prions to amyloidogenic proteins (*Zhu et al., 2003*; *Kundel et al., 2018*; *Ruschak and Miranker, 2007*; *Meisl et al., 2014*; *Cohen et al., 2013*); key examples of such secondary nucleation processes include fibril fragmentation, lateral branching and surface-catalyzed secondary nucleation.

In the presence of a liquid compartment, irreversible protein aggregation of fibrillar structures occurs within each phase as a consequence of both primary and secondary nucleation, and growth of aggregates via their ends, each event occurring with rate constants $k_1$, $k_2$, and $k_+$ (*Michaels and Knowles, 2014*; *Michaels et al., 2016*; *Arosio et al., 2016*; *Michaels et al., 2018*). We have seen that the key term in our model is the difference between the monomer concentration inside and outside of the compartment which leads to the diffusive flux of monomers $J^\alpha$ between the phases (*Equation 3a*), which connects the effects of phase separation and protein aggregation. The coupled equations describing protein aggregation kinetics in both phases can be written as

$$\frac{\mathrm{d}c_{\mathrm{a}}^\alpha(\mathrm{t})}{\mathrm{dt}} = k_1 M_{\mathrm{m}}^\alpha(t)^{n_1} + k_2 M_{\mathrm{m}}^\alpha(t)^{n_2} M_{\mathrm{a}}^\alpha(t), \tag{3b}$$

$$\frac{\mathrm{d}M_{\mathrm{a}}^\alpha(\mathrm{t})}{\mathrm{dt}} = 2k_+ M_{\mathrm{m}}^\alpha(t) c_{\mathrm{a}}^\alpha(t), \tag{3c}$$

$$\frac{\mathrm{d}M_{\mathrm{m}}^\alpha(\mathrm{t})}{\mathrm{dt}} = -2k_+ M_{\mathrm{m}}^\alpha(t) c_{\mathrm{a}}^\alpha(t) + \frac{J^\alpha}{V^\alpha}. \tag{3d}$$

Here, *Equation (3b)* describes the rate of formation of new fibrils in each compartment ($\alpha = \mathrm{I}, \mathrm{II}$) through primary nucleation, fragmentation or surface catalyzed secondary nucleation. In the case of primary nucleation, the rate of formation of new aggregates depends solely on the concentration of monomers, where the reaction order $n_1$ describes the concentration dependence of nucleation. For secondary processes, including fragmentation and surface-catalyzed secondary nucleation, the rate

of formation of new aggregates is proportional to the aggregate mass concentration; the dependence of the rate on the monomer concentration is described by the reaction order $n_2$ (the case $n_2 = 0$ corresponding to fragmentation). Note that both primary and secondary nucleation of aggregates are non-classical, multi-step nucleation processes; hence, the reaction orders $n_1$ and $n_2$ do not necessarily correlate to the physical size of nuclei (*Šarić et al., 2016*). *Equation (3c)* captures the build-up of aggregate mass within each compartment due to elongation of existing aggregates, which occurs by monomer addition at their ends. Finally, *Equation (3d)* models the population balance of monomers in each compartment as a result of two effects: (i) monomer depletion due to aggregate growth (see *Equation (3c)*) and (ii) the monomer flux between compartments I and II; this flux is given by *Equation (3a)* and ensures that partitioning is maintained close to the monomer partitioning factor $\Gamma$.

While the monomer partitioning factor $\Gamma$ (*Equation 1*) governs the constant ratio of the time dependent concentrations in compartment I and II, the partitioning degree $\xi$ (*Equation 2*) determines how the total monomer concentration, which decays over time as a result of aggregation, is split between the two compartments at any time point during the kinetics of aggregation. As we will see, both parameters will be crucial in controlling the degree of aggregate partitioning into the compartments.

### Irreversible aggregation in the presence of phase separated compartments

To understand how protein aggregation kinetics couples to the two phase separated compartments in terms of the physical parameters $\Gamma$ and $\xi$, we constructed explicit analytical solutions to the set of non-linear kinetic *Equation (3)* by exploiting an analogy to classical mechanics (*Michaels et al., 2016* and Appendix B for details of the calculations), and compared these with numerical solutions of (*Equation 3*).

### Monomer partitioning affects nucleation and growth of aggregates between the compartments

In the limit of fast monomer diffusion, the aggregation kinetics in each compartment is controlled by a set of effective rate parameters. The relative magnitude of these effective rates between compartment I and II at early times scales with the monomer partitioning as $\Gamma^{n_1}$, while at late times, the corresponding ratio of these rates scales with $\Gamma^{(n_2+1)/2}$ (see Appendix 3, *Equation (39)* and *Equation (40)*). Thus, the aggregate growth inside compartment I is faster than in compartment II if there is enrichment of monomers in the condensed phase ($\Gamma > 1$). Moreover, the relative magnitudes of growth rate at early times solely depends on the reaction order of primary nucleation, $n_1$, while at late times, relative growth is determined by the reaction order of secondary nucleation, $n_2$.

### Phase separated compartments mediate a positive feedback for aggregate growth

This difference in growth rates between the phases can be qualitatively explained by the rapid preference of monomers to recover phase equilibrium (*Figure 3(a)*). The enhanced monomer concentration in compartment I causes aggregates to nucleate first inside compartment I. As a consequence, elongation of aggregates is more pronounced inside compartment I leading to a stronger consumption of monomers. This difference in monomer consumption between the compartments couples to the flux (*Equation 3*), which forces more monomers to diffuse into compartment I to maintain partitioning equilibrium, even as aggregates grow. This positive feedback mechanism in compartment I is accompanied by negative feedback for compartment II, which continuously loses monomers leading to a slowing down of the aggregation kinetics outside. Thus, the coupling between the aggregation kinetics and phase separation, mediated by diffusion of monomers (*Equation 3*), is key to determine aggregate enrichment/depletion in each phase.

### Positive feedback for aggregate growth causes strong aggregate partitioning

To understand this feedback mechanism, we study the time evolution of the aggregate concentration inside each phase, $c_a^I(t)$ and $c_a^{II}(t)$ (*Figure 3(b)*). The first aggregates are initiated by primary

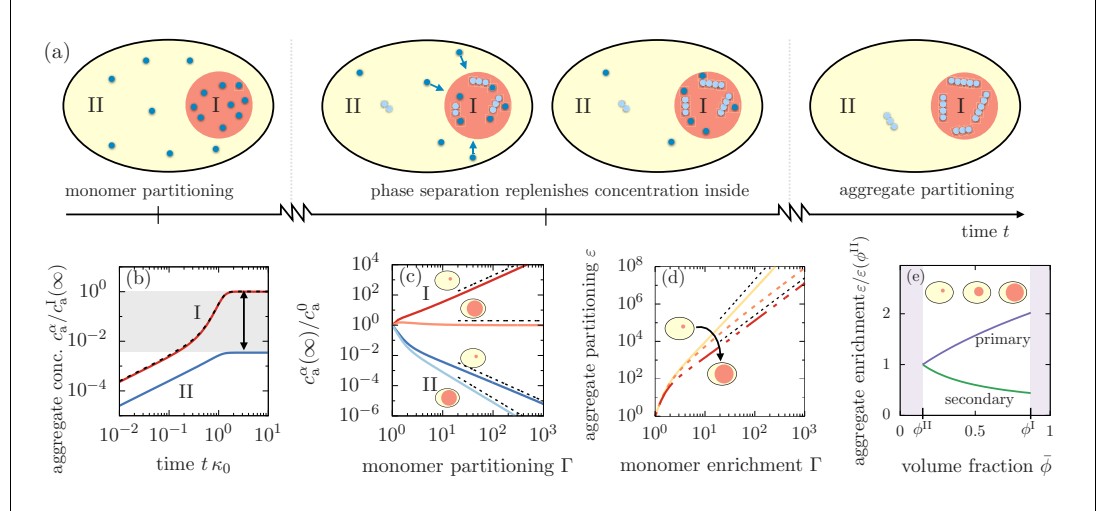

**Figure 3.** Segregation of aggregates into compartment I via positive feedback mediated by phase separation. (a) Sketch of aggregation kinetics inside the two compartments I and II. Left: Initially, monomers get enriched on a short diffusive time scales due to the partitioning mediated by the phase separated compartments (*Equation 1*). Center: Monomers slowly aggregate. More aggregates nucleate and grow in compartment I due to the initial partitioning of monomers. This pronounced, initial aggregation causes a continuous monomer flux into compartment I, further promoting aggregation (positive feedback indicated by arrows). Right: Partitioning of monomers together with the positive feedback can cause a very pronounced accumulation of aggregates relative to compartment II. (b) Aggregate concentration $c_a^\alpha(t)$ as a function of time $t$ obtained from solving numerically and analytically *Equation 3* actually confirms that aggregates can enrich by several orders of magnitude. (c) The asymptotic concentrations $c_a^I(\infty)$ and $c_a^{II}(\infty)$ inside each of the compartment inversely scale for small compartments, while for large compartment I, aggregate enrichment therein vanishes while depletion inside compartment II is dominated by primary nucleation. The asymptotic concentration in the absence of monomer partitioning, $\Gamma = 1$, is denoted as $c_a^{tot}|_{\phi^I=\phi^{II}}$. Dashed line are the scalings given in the the main text. Parameters: $n_1 = n_2 = 2$. (d) Partitioning factor $\varepsilon$ of aggregates inside compartment I as a function of monomer partitioning $\Gamma$ can reach very large values. The behavior switches from secondary nucleation dominated increase at small compartment I volumes to primary dominated growth at large volumes. Dashed line are the scalings given in *Equation (6)*. (e) The slope of the partitioning factor as a function of mean volume fraction $\bar{\phi}$, equivalently speaking, volume of compartment I, changes its sign when partitioning is dominated by primary ($n_1 = 2, n_2 = 0$) or secondary nucleation ($n_1 = 2, n_2 = 2$). Parameters: (b,e) $\Gamma = 3$ consistent with weak interactions.

DOI: https://doi.org/10.7554/eLife.42315.004

nucleation and solely determined by the monomer concentration. Because monomer concentrations in the compartments are slaved due to the rapid flux that maintains partitioning equilibrium, the time evolution of the aggregate concentrations in the early regime of the aggregation kinetics are slaved as well, following $c_a^I(t)/c_a^{II}(t) \propto \Gamma^{n_1}$. When aggregates start consuming monomers via elongation, the flux of monomers from compartment II to I causes a saturation of the aggregate concentration outside the compartment II, while the concentration of aggregates in compartment I increases significantly. This rapid increase of growth is facilitated by the continuous influx of monomers (positive feedback). As monomers get depleted in the entire system the growth of aggregates also saturates in compartment I. Most importantly, the resulting asymptotic concentrations at large time scales, $c_a^I(\infty)$ and $c_a^{II}(\infty)$, can differ by several orders of magnitude, even for modest values of $\Gamma$ corresponding to weak relative interactions.

## Enrichment and depletion relative to homogeneous aggregation is determined by the reaction orders

To elucidate the impact of the reaction orders on the aggregation kinetics, we first consider the enrichment of aggregates relative to the case of homogeneous aggregation, that is for $\Gamma = 1$. For large values of monomer partitioning, the asymptotic concentrations in compartments I and II at large times relative to the homogeneous aggregate concentration $c_a^{tot}|_{\phi^I=\phi^{II}}$ at large times read

$$\frac{c_{\mathrm{a}}^{\mathrm{II}}(\infty)}{w \sim c_{\mathrm{a}}^{\mathrm{tot}}|_{\phi^{\mathrm{I}}=\phi^{\mathrm{II}}}} \simeq \xi(\bar{\phi})^{n_1 - \frac{n_2+1}{2}} \, \Gamma^{-\frac{n_2+1}{2}}, \tag{4}$$

$$\frac{c_{\mathrm{a}}^{\mathrm{I}}(\infty)}{c_{\mathrm{a}}^{\mathrm{tot}}|_{\phi^{\mathrm{I}}=\phi^{\mathrm{II}}}} \simeq \left(\xi(\bar{\phi}) \, \Gamma\right)^{\frac{n_2+1}{2}}, \tag{5}$$

where $w$ is a dimensionless numerical prefactor (Appendix 3, *Equation 52*). We see that for a large monomer partitioning factor $\Gamma$, the partitioning of aggregates inside compartment I gets more pronounced, while aggregates in compartment II are more depleted relative to the homogeneous case (*Figure 3(c)*). Most importantly, the value of the terminal values of aggregate concentrations for given monomer partitioning factor are controlled by the reaction orders for primary and secondary nucleation, $n_1$ and $n_2$. The role of $n_1$ and $n_2$ results directly from the interplay between aggregate growth and nucleation and their dependence on the monomer concentration.

## Aggregate concentration in the compartments is controlled by compartment volume

Having understood the role of the monomer partitioning factor $\Gamma$ in aggregation kinetics, we now turn to how the asymptotic concentrations of aggregates in each compartment depend on the volume of the compartments. The dependence on compartment volume is given the partition degree $\xi(\bar{\phi})$. From *Equation 2*, we see that the partition degree $\xi(\bar{\phi}) \in [1, \Gamma^{-1}]$, where the value of one is relevant for small compartments (*Figure 2(b)*). Following *Equations (4) and (5)*, we see that for a small volume of compartment I, enrichment and depletion exhibit an inverse scaling, i.e. $c_{\mathrm{a}}^{\mathrm{I}}(\infty) \propto \left(c_{\mathrm{a}}^{\mathrm{II}}(\infty)\right)^{-1} \propto \Gamma^{\frac{n_2+1}{2}}$, which is solely dependent on the reaction order for secondary nucleation. Contrariwise, when the volume of compartment I is large, enrichment of aggregates inside I vanishes, while depletion inside compartment II then solely depends on the reaction order for primary nucleation, $c_{\mathrm{a}}^{\mathrm{II}}(\infty) \propto \Gamma^{-n_1}$.

This switch between aggregate partitioning governed by secondary nucleation, to a partitioning solely determined by primary nucleation, arises from primary nucleation events occurring first inside compartment I due to a higher monomer concentration ($\Gamma > 1$). Once the first aggregates have formed via primary nucleation inside compartment I, small and large compartments behave fundamentally differently. If compartment I is small, only a few aggregates can form via primary nucleation due to the small compartment size. As aggregates begin to grow earlier in compartment I, the unbalance of monomers causes a flux from II to I. As a consequence of this continuous flux, the secondary nucleation events quickly overwhelm primary nucleation events inside compartment I, while secondary nucleation is suppressed in compartment II. However, if compartment I is large, the aggregation kinetics is similar to that for a homogeneous system because the monomer mass concentration is very close to the total monomer mass in the system and there is only a negligible amount of monomers entering from compartment II. Additionally, in the smaller compartment II where aggregates grow via primary nucleation, the coupling flux continuously removes monomers suppressing primary nucleation. Since compartment I is large, it shows little or no enrichment of aggregates relative to the homogeneous case while inside the small compartment II, aggregates are depleted determined by the lack of primary nucleation events relative to the homogeneous case.

## Changes in compartment volume switch the driving mechanism for aggregate partitioning

To quantify the switch in aggregate partitioning as a function of compartment volume, we define the asymptotic aggregate partitioning ratio

$$\varepsilon(\bar{\phi}) = \frac{c_{\mathrm{a}}^{\mathrm{I}}(\infty)}{c_{\mathrm{a}}^{\mathrm{II}}(\infty)} \propto \xi(\bar{\phi})^{n_2 - n_1 + 1} \, \Gamma^{n_2+1}. \tag{6}$$

As the compartment volume enters the partitioning factor $\varepsilon(\bar{\phi})$ solely via the partition degree $\xi(\bar{\phi})$, the sign of $n_2 - n_1 + 1$ determines whether larger or smaller compartments lead to a larger partitioning (*Figure 3(e)*). Indeed, we find that the slope of the partitioning factor scales as

$\varepsilon(\bar{\phi})' \propto (n_1 - n_2 - 1)$. Thus, for $n_1 > n_2 + 1$, increasing the compartment volume by increasing the amount of $A$-material $\bar{\phi}$ causes a larger relative partitioning. Conversely, for $n_1 < n_2 + 1$, larger partitioning can be found for smaller compartment sizes. Consistently, if the nucleation coefficients obey $n_2 = n_1 - 1$, compartment volume has no impact on the partitioning factor $\varepsilon$.

This qualitative switch in the mechanism for aggregate partitioning raises the question which systems favor large or small compartment volumes in order to maximize aggregate partitioning $\varepsilon(\bar{\phi})$. *Figure 4* depicts the regimes in terms of the reaction orders characterizing primary and secondary nucleation, $n_1$ and $n_2$, for which the maximal aggregate partitioning corresponds to smaller and larger compartment volumes. This prediction can be related to specific aggregating systems for which the values of the reaction orders $n_1$ and $n_2$ have been experimentally determined (References see caption of *Figure 4*). Using these values for the reaction orders, our model predicts that largest partitioning is obtained for large compartments in systems of aggregating tau and yeast prion Ure2p. These two examples belong primarily to the class of systems where the mechanism responsible for the formation of new aggregates in the late stage is fragmentation which has a zero secondary reaction order, $n_2 = 0$ (i.e. nucleation is monomer independent). For non-fragmenting systems with $n_2 > 0$, our model predicts different scenarios for aggregating systems: largest aggregate partitioning for large compartment volumes occurs in the case branching systems, such as actin in the presence of the complex Arp2/3, as well as systems proliferating through monomer dependent secondary nucleation with $n_2 < n_1 - 1$, such as the Islet Amyloid Polypeptide (IAPP). In contrast, largest

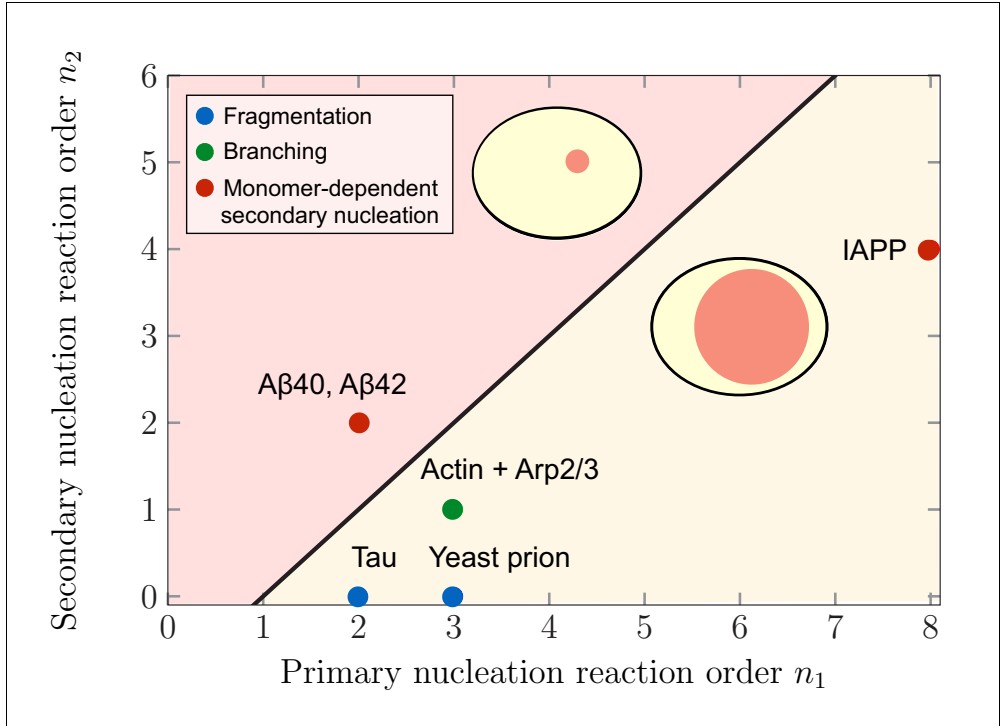

**Figure 4.** Theoretical predictions of maximal aggregate partitioning for various aggregating systems. Our predictions are summarized by a phase diagram depicting that aggregating systems characterized by different reaction orders for primary and secondary nucleation, $n_1$ and $n_2$, show maximal aggregate partitioning for large or small compartments, respectively. The two regions where either large or small compartments lead to a larger partitioning of aggregates is separated by the line $n_2 = n_1 - 1$ determined from *Equation 6*. For $n_2 > n_1 - 1$ small compartments lead to larger aggregate partitioning, while for $n_2 < n_1 - 1$, larger compartments are beneficial. To illustrate which scenario might apply to which kind of aggregating system, we indicate the measured values of the primary and secondary reaction orders for a range of systems propagating through fragmentation (blue), lateral branching (green) or monomer-dependent secondary nucleation (red): Tau (*Kundel et al., 2018*), yeast prion Ure2p (*Zhu et al., 2003*), IAPP (*Ruschak and Miranker, 2007*), Amyloid-β40 (for monomer concentrations below 5 µM) (*Meisl et al., 2014*), Amyloid-β42 (*Cohen et al., 2013*).
DOI: https://doi.org/10.7554/eLife.42315.005

aggregate partitioning is reached for small compartments in the case of the 40- and 42-residue forms of Amyloid-β peptide (Aβ40 and Aβ42).

## Compartment volume and monomer partitioning control the total amount of aggregates

Our results have demonstrated that aggregates can be effectively partitioned inside liquid-like compartments, raising the question: can compartments also control the total amount of aggregates or their average size? To test this possibility, we compute the difference between the total amount of aggregates formed in the presence of liquid compartments, $c_a^{tot} = \left(c_a^I(\infty)V^I + c_a^{II}(\infty)V^{II}\right)/V$, compared to the number of aggregates formed in the homogeneous system without compartments, $c_a^{tot}|_{\phi^I=\phi^{II}}$. The homogeneous case can be studied by considering equal compositions of both compartments, that is $\phi^I = \phi^{II}$. This difference between the homogeneous case and the case with compartments be quantified by introducing the relative asymptotic aggregate concentration, $\mathcal{C}(\bar{\phi}, \Gamma) = (c_a^{tot} - c_a^{tot}|_{\phi^I=\phi^{II}})/c_a^{tot}|_{\phi^I=\phi^{II}}$, which is positive for an increased pool of aggregates, and negative for a lowered pool of aggregates relative to the homogeneous state. We find that compartments can affect the total number of aggregates relative to the homogeneous system depending on the relative values of the reaction orders for secondary nucleation and aggregate growth, the value of monomer partitioning $\Gamma$ and the amount of compartment material $\bar{\phi}$ that in turn regulates compartment size $V^I$. In particular, for reaction orders $n_2 < 1$, the presence of the liquid droplet *always* reduces the total amount of aggregates formed relative to the homogeneous system for all values of $\bar{\phi}$ and $\Gamma$ and thereby *always* leading to larger aggregates (*Figure 5(a)*). However, for $n_2 > 1$, we find a different behaviour. For low partitioning factors $\Gamma$, the presence of liquid compartments decreases the total number of aggregates, corresponding to a larger average aggregate size, while for larger values of $\Gamma$, more and thereby shorter aggregates form compared to the homogeneous system (*Figure 5(b)*). This behavior is also affected by compartment volume; the corresponding boundary in the $\bar{\phi}$-$\Gamma$ diagram separates these two regimes corresponding to more but smaller or less but larger aggregates (*Figure 5(c)*). The role of the reaction order for secondary nucleation $n_2$ on total aggregate concentration and average size can be explained as follows. In a homogeneous system proliferating through secondary nucleation pathways, the average aggregate size in the saturating regime of the aggregation kinetics at long times scales as $\sqrt{k_+/k_2}\,[M_m^{tot}]^{(1-n_2)/2}$ (*Michaels et al., 2015*). Larger values of $\Gamma$ lead to an increase of monomers in the compartment, favouring both secondary nucleation and aggregate growth by elongation inside the compartment. If $n_2 > 1$, the rate of secondary nucleation is increased by $\Gamma$ more than elongation, which results in more numerous aggregates and hence shorter aggregates due to the limiting and fixed amount of total monomer mass in the system. The opposite trend is observed when $n_2 \leq 1$. In summary, a strong partitioning of aggregates inside compartments caused by a strong monomer partitioning (large $\Gamma$) is accompanied by an increase of the total number of aggregates in the system in the presence of secondary nucleation, while in the absence of secondary nucleation, the total amount of aggregates decreases.

## Discussion

By combining the theories of irreversible protein aggregation kinetics and phase separation, we have shown how liquid compartments can control the position and the total amount of aggregates. The coupling of slow aggregation and rapid phase separation leads to a mechanism whereby even a weak partitioning of monomers is amplified into a relatively large accumulation of aggregates in the compartment. Such partitioning of aggregates is a non-equilibrium effect and thereby not only determined by the phase separation parameters relevant at equilibrium (monomer partitioning $\Gamma$ and partitioning degree $\xi$) but in addition, it depends on kinetic parameters characterizing the aggregation kinetics (e.g. reaction orders $n_1$ and $n_2$ for primary and secondary nucleation). However, several other effects may influence or limit the resulting degree of aggregate partitioning.

### Model validity

In our model, we have considered the case that monomers and aggregates do not affect phase separation and phase separation is driven by the competition between the entropic tendency to mix and interactions favoring demixing. Future work could be devoted to extending our model by a

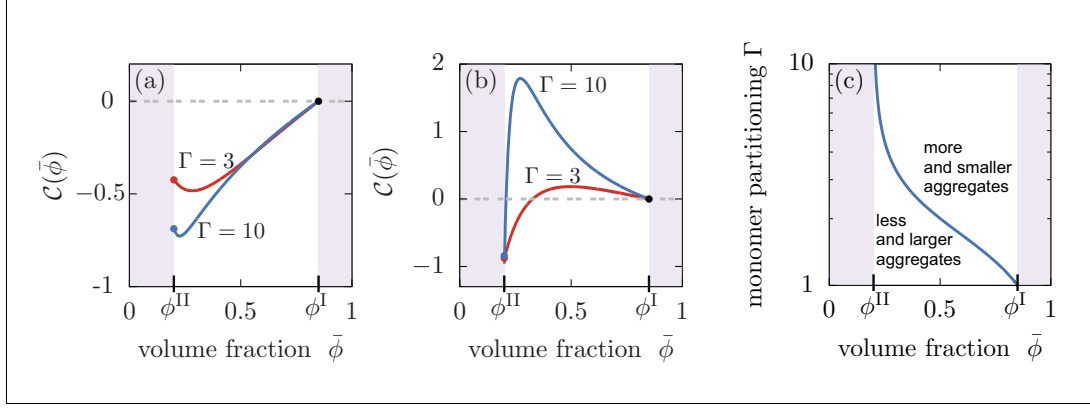

**Figure 5.** Compartments can change the total aggregate concentration compared to the homogeneous state without compartments. (a,b) Relative asymptotic aggregate concentration $\mathcal{C}(\bar{\phi}, \Gamma) = (c_{\mathrm{a}}^{\mathrm{tot}} - c_{\mathrm{a}}^{\mathrm{tot}}|_{\phi^{\mathrm{I}}=\phi^{\mathrm{II}}})/c_{\mathrm{a}}^{\mathrm{tot}}|_{\phi^{\mathrm{I}}=\phi^{\mathrm{II}}}$ as a function of volume fraction of the compartment material $\bar{\phi}$ (connected to compartment volume $V^{\mathrm{I}}(\bar{\phi}) = V(\bar{\phi} - \phi^{\mathrm{II}})/(\phi^{\mathrm{I}} - \phi^{\mathrm{II}})$), where $c_{\mathrm{a}}^{\mathrm{tot}}|_{\phi^{\mathrm{I}}=\phi^{\mathrm{II}}}$ is the concentration of the homogeneous state in the absence of compartments. (a) For secondary reaction order $n_2 < 1$, the total amount of aggregates is decreased compared to the case without compartments for all values of monomer partitioning $\Gamma$ and compartment material volume fractions $\bar{\phi}$ and compartment volumes $V^I$. (b) However, for $n_2 > 1$, the total amount of aggregates is either increased or decreased relative to the homogeneous state. (c) Depending on the value of the monomer partitioning $\Gamma$, compartments either lead to more but shorter aggregates (large $\Gamma$, larger volume controlled by $\bar{\phi}$) or less but larger aggregates Parameters: (a) $n_1 = 2$, $n_2 = 0$; (b,c) $n_1 = 2$, $n_2 = 3$.
DOI: https://doi.org/10.7554/eLife.42315.006

coupling between aggregates and the liquid compartment or by entropically driven phase separation, relevant for the assembly of coacervates (*Overbeek and Voorn, 1957*) or mixtures with depletion interactions. Moreover, our model is restricted to time scales when aggregates hardly diffuse and monomer diffusion is not affected by rheological properties of the aggregates. The observed strong aggregate partitioning may diminish if aggregates significantly diffuse, or if they slow down diffusion of monomers. Furthermore, we have focussed on the case where both phases have the same reaction rates of aggregation. This assumption may be inaccurate for protein-rich phases (*Wei et al., 2017*) but can be scrutinized using our model (see Appendix 3). We find that due to the power law dependence of the monomer partitioning $\Gamma$ (*Equation 6*) differences in aggregation rates must be very large to significantly affect the partitioning of aggregates. A lowered partitioning of aggregates could be caused by the coarsening dynamics of many droplets (*Ostwald, 1897*; *Lifshitz and Slyozov, 1961*; *Bray, 1994*). While coarsening via coalescence would not affect our results at all because aggregates remain confined inside the droplets, dissolving droplets undergoing Ostwald ripening would diminish the degree of aggregate partitioning. However, because the aggregation kinetics varies with compartment size, aggregates in droplets of different size may compete about monomers. This non-equilibrium competition could cause accumulation of more aggregates either in smaller or larger compartments. Overall, for systems where partitioning of monomers is fast relative to the aggregation kinetics, the mechanism underlying the strong partitioning of aggregates proposed in this study could be relevant for several phenomena in living cells. It could have impact on strategies of drug design or serve as a principle to speed up irreversible chemical reactions and can be tested experimentally.

## In-vitro realization

Our quantitative predictions of strong aggregate enrichment inside a liquid-like compartment (*Figure 3 (b–e)* and *Figure 4*) are experimentally testable using recently developed bulk and microfluidic assays. For example, synthetic liquid biocompartments of tuneable size and composition can be used to locally affect reaction rates and partition proteins (*Faltova et al., 2018*) and thereby represent attractive platforms to investigate the partitioning and aggregation of different amyloidogenic peptides and proteins, including Amyloid-β. These synthetic compartments are highly

flexible and allow to validate the effect of several parameters predicted in this work. For instance, the monomer partitioning factor $\Gamma$ could be varied in vitro by changing the degree of phase separation (*Equation (1)* and *Figure 2*) or by conjugating the proteins with specific sequences capable of tuning recruitment into the liquid compartments (*Faltova et al., 2018*). Moreover, the compartment volumes can be adjusted by the initial supersaturation via changes in temperatures, which affect the kinetic rate constants only weakly (*Cohen et al., 2018*). Measuring the concentration or size of aggregates inside and outside of the compartment by epi-fluorescence spectroscopy as a function of time and parameters such as the partitioning factor and compartment volume will allow for tracking aggregate enrichment as a function of compartment volume and test both the scaling predictions and the crossover of the scaling exponent from $n_2 + 1$ at small volume to $n_1$ at large volumes. It would be particularly interesting to test this prediction for different amyloid-forming protein systems or for varying the reaction orders $n_1$ and $n_2$ by adjusting the total amount of monomers (*Meisl et al., 2014*); see *Figure 4*.

## In vivo relevance and implications for drug design

Our model may already provide a framework to explain the phenomena of aggregate partitioning inside living cells. An example of such phenomena could be the partitioning of pericentriolar material into centrosomes (*Zwicker et al., 2014*) and the spatial organization of aggregates inside stress granules (*Molliex et al., 2015*; *Mateju et al., 2017*). The propensity of aggregates to solidify the compartment as reported in *Mateju et al. (2017)* could be accounted for in our model through a gel-sol transition (*Stockmayer, 1943*; *Harmon et al., 2017*). Including the solidification induced by aggregates could lead to additional volume changes of the compartment which in turn may affect the aggregation kinetics. Furthermore, the enrichment of toxic aggregates inside liquid compartments may trigger new directions for drug design against aberrant protein aggregation. Our results suggest to design drugs not only with respect to their ability to interfere with the aggregation kinetics (*Arosio et al., 2014*) but also with respect to their partitioning properties into the liquid compartments. This strategy is reminiscent of quantifying the potency of low-molecular weighted anesthetics via the Meyer-Overton correlation based on solubility of the anaesthetics in oil (*Meyer, 1937*; *Franks and Lieb, 1978*).

## General speed-up mechanism for chemical reactions

The reported feedback mechanism of aggregate growth mediated by liquid compartments may represent a general principle to spatially confine and speed up other irreversible chemical processes or to control aggregate amount and average size. Examples may include precipitation of proteins or polymerization kinetics of actin and microtubules (see also *Figure 4*). Indeed, a speed up of the chemical reactions could be expected due to the increased concentration of educts inside the liquid compartments. Thus, liquid compartments are ideal biomolecular microreactors that enrich the amount of products by dynamically exchanging reactants with their surroundings.

## Acknowledgements

CAW thanks the German Research Foundation (DFG) for financial support and TCTM acknowledges the support from the Swiss National Science foundation.

## Additional information

### Funding

| Funder | Author |
| --- | --- |
| Deutsche Forschungsge-meinschaft | Christoph Weber |
| Swiss National Science Foundation | Thomas Michaels |
| MacArthur Foundation | L Mahadevan |

The funders had no role in study design, data collection and interpretation, or the decision to submit the work for publication.

## Author contributions

Christoph Weber, Thomas Michaels, Conceptualization, Formal analysis, Validation, Investigation, Writing—original draft, Writing—review and editing; L Mahadevan, Conceptualization, Formal Analysis, Validation, Supervision, Funding acquisition, Project administration, Writing—review and editing

## Author ORCIDs

L Mahadevan (iD) https://orcid.org/0000-0002-5114-0519

## Decision letter and Author response

Decision letter https://doi.org/10.7554/eLife.42315.013
Author response https://doi.org/10.7554/eLife.42315.014

# Additional files

## Supplementary files

• Transparent reporting form
DOI: https://doi.org/10.7554/eLife.42315.007

## Data availability

All data generated or analysed during this study are included in the manuscript and supporting files.

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

## Appendix 1

DOI: https://doi.org/10.7554/eLife.42315.008

# Partition coefficient for dilute monomers at equilibrium

We consider phase separation of an incompressible, ternary mixture composed of monomers, component $A$ and $B$ (we neglect the interactions between the aggregates and phase separation for simplicity) described by the following Flory-Huggins free energy density (*Flory, 1942*; *Huggins, 1942*)

$$f = k_B T \left[ \frac{\phi_A}{\nu_A} \ln(\phi_A) + \frac{\phi_B}{\nu_B} \ln(\phi_B) + c_m \ln(\nu_m c_m) + \Lambda \phi_A \phi_B + c_m \nu_m (\Lambda_{mA} \phi_A + \Lambda_{mB} \phi_B) \right], \quad (7)$$

where the logarithmic terms correspond to the mixing entropy. The interactions between $A$ and $B$ are described by the parameter $\Lambda$ while the interactions with the monomers are characterized by $\Lambda_{mi}$, $i = A, B$. These interaction parameters have the unit $[1/volume]$. Here we define a dimensionless interaction parameters by writing $\Lambda = \chi/\nu$ and $\Lambda_{mi} = \chi_{mi}/\nu$ with $\nu = \nu_A$. Moreover, for simplicity, we consider equal molecular volumes of $A$ and $B$, $\nu = \nu_B$. Thus we arrive at

$$f = \frac{k_B T}{\nu} [\phi_A \ln(\phi_A) + \phi_B \ln(\phi_B) + \nu c_m \ln(\nu_m c_m) + \chi \phi_A \phi_B + c_m \nu_m (\chi_{mA} \phi_A + \chi_{mB} \phi_B)]. \quad (8)$$

With $\phi_B = 1 - \phi_A - \nu_m c_m$ and monomers being dilute, $\nu_m c_m \ll 1$, we can expand $f$ in $\nu_m c_m$ up to the first order:

$$f(\phi, c_m) \simeq \frac{k_B T}{\nu} \{ \phi \ln(\phi) + (1 - \phi) \ln(1 - \phi) + \chi \phi (1 - \phi)$$

$$+ \nu_m c_m ((\nu/\nu_m) \ln(\nu_m c_m) - \ln(1 - \phi) + (\chi_{mA} - \chi_{mB} - \chi) \phi + \chi_{mB} - 1) \}, \quad (9)$$

where we defined $\phi_A \equiv \phi$ and neglected all terms of order $\mathcal{O}\left[(\nu_m c_m)^2\right]$. The corresponding chemical potentials read

$$\tilde{\mu}(\phi, c_m) = \nu \frac{\partial f}{\partial \phi} = k_B T [\ln \phi - \ln(1 - \phi)$$

$$+ \chi (1 - 2\phi) + c_m \nu_m \left( \chi_{mA} - \chi - \chi_{mB} + \frac{1}{1 - \phi} \right)], \quad (10a)$$

$$\mu_m(\phi, c_m) = \frac{\partial f}{\partial c_m} = k_B T \{ \ln(\nu_m c_m) + 1$$

$$+ \frac{\nu_m}{\nu} \ln(1 - \phi) + \frac{\nu_m}{\nu} [(\chi_{mA} - \chi) \phi + \chi_{mB} (1 - \phi) - 1] \}. \quad (10b)$$

At equilibrium the chemical potentials of each component inside (I) and outside (II) the compartment are balanced leading to relationships between the concentration values inside and outside. Specifically, if phase separation equilibrium is reached the following conditions are fulfilled

$$\tilde{\mu}^I(\phi^I, c_m^I) = \tilde{\mu}^{II}(\phi^{II}, c_m^{II}), \quad (11a)$$

$$\mu_m^I(\phi^I, c_m^I) = \mu_m^{II}(\phi^{II}, c_m^{II}). \quad (11b)$$

The relations above allow to calculate the equilibrium concentration in each phase, for component $A$, $\phi^I$ and $\phi^{II}$, and the monomers, $c_m^I$ and $c_m^{II}$. An analytic result of the equilibrium concentrations is very difficult to obtain. However, we can focus on the leading contributions for the balance of the chemical potentials inside and outside taking advantage that monomers are dilute and thereby obtain an approximation for the equilibrium values inside and outside.

Considering that the dimensionless interaction parameters $\chi$ are all of $\mathcal{O}(1)$, the impact of the dilute monomers on the phase equilibrium between $A$ and $B$ is negligible and we can approximate

$$\tilde{\mu}(\phi, c_{\mathrm{m}}) \simeq k_B T\left[\ln\left(\frac{\phi}{1-\phi}\right) + \chi(1-2\phi)\right]. \qquad (12)$$

The resulting chemical potential **Equation (12)** simply corresponds to the chemical potential of a binary, incompressible Flory-Huggins mixture. From the equilibrium condition **Equation (11a)**, we can calculate the binodal line described by the condition

$$\chi \simeq \frac{\ln\left(\phi/(1-\phi)\right)}{2\phi - 1}, \qquad (13)$$

which solely depends on the interaction parameters between $A$ and $B$, $\chi$. For fixed interaction parameters $\chi$, the binodal gives the equilibrium volume fractions inside and outside the compartment, $\phi^{\mathrm{I}}$ and $\phi^{\mathrm{II}}$. For finite sized compartments, these equilibrium volume fractions are slightly increased due to the Laplace pressure. This increase is described by the Gibbs-Thomson relationships, $\phi_{eq}^{\alpha} = \phi^{\alpha}(1 + \ell_{\gamma}^{\alpha}/R)$, where $\ell_{\gamma}^{\alpha}$ denotes the capillary length for each phase. For strong phase separation of compartment I, $\phi^{\mathrm{I}} \gg \phi^{\mathrm{II}}$, the capillary lengths obey $\ell_{\gamma}^{\mathrm{I}} \ll \ell_{\gamma}^{\mathrm{II}}$ (**Weber et al., 2018**). Moreover, the capillary length in the minority phase II can be estimated by $\ell_{\gamma}^{\mathrm{II}} \simeq 2\gamma\nu/(\phi^{\mathrm{I}}k_B T)$, which is typically closer to the molecular size than the size of micron-sized compartments of radius $R$. Thus, for compartments significantly exceeding the size of the molecules we can neglect Laplace and surface tension effects on the equilibrium volume fraction and approximate $\phi_{eq}^{\alpha} \simeq \phi^{\alpha}$ in the following.

By means of equilibrium condition **Equation (11b)**, we can calculate the impact of the phase separated compartment on the monomer distribution leading to the monomer partitioning factor

$$\Gamma \equiv \frac{c_{\mathrm{m}}^{\mathrm{I}}}{c_{\mathrm{m}}^{\mathrm{II}}} \simeq \exp\left[\frac{\nu_{\mathrm{m}}}{\nu}\Delta\chi\left(\phi^{\mathrm{I}} - \phi^{\mathrm{II}}\right)\right], \qquad (14)$$

where the relative interaction strength reads $\Delta\chi = \left(\chi_{B,m} - \chi_{A,m}\right)$. Thus, there is partitioning of monomers into the condensed phase ($\Gamma > 1$) if monomers favor the presence of $A$ relative to $B$, that is $\chi_{A,\mathrm{m}} < \chi_{B,\mathrm{m}}$.

## Appendix 2

DOI: https://doi.org/10.7554/eLife.42315.008

### Inter-compartment flux of monomers close to equilibrium

This flux can be calculated for a maintained concentration difference $\phi^{\mathrm{I}} - \phi^{\mathrm{II}}$ using the chemical potential for the monomers $\mu_{\mathrm{m}}$ (**Equation 10b**). To this end, let us consider that the compartment is spherical of radius $R$. Perturbing the concentrations in both phases may lead to an unbalance of the chemical potential and thus a flux between the phases. The net radial flux of monomers $J = J_{\mathrm{r}}^{\mathrm{I}} - J_{\mathrm{r}}^{\mathrm{II}}$ [unit: mass per time] arises from a flux contribution of each phase through the interface, where the radial flux in each phase reads

$$J_{\mathrm{r}}^{\alpha} = 4\pi R^2 \boldsymbol{e}_r \cdot \boldsymbol{j}^{\alpha}|_R\,, \tag{15}$$

and $\boldsymbol{e}_r$ is the radial unit vector pointing normal to the spherical interface. The local fluxes at the interface, $\boldsymbol{j}^{\alpha}|_R$, can be calculated in the limit where the aggregation kinetics is slow compared to diffusion in each phase. In this limit we can use a quasi static approximation, $\nabla \cdot \left(\xi^{\alpha}\nabla\mu_m^{\alpha}(r)\right) \simeq 0$. For a constant mobility coefficient in each phase we can solve a Laplace equation for the chemical potential, $\nabla^2\mu_m^{\alpha}(r) \simeq 0$, with the following boundary conditions for the chemical potentials: $\mu_m^{\alpha}$ deeply inside each phase and $\mu_m^{\alpha}|_R$ at the interface $r = R$. For simplicity, we discuss the case of a small compartment I with a homogeneous chemical potential inside, thus $\mu_{\mathrm{m}}^{\mathrm{I}}(r) = \mu_{\mathrm{m}}^{\mathrm{I}}$ is constant and equal to the chemical potential right inside the interface $\mu_{\mathrm{m}}^{\mathrm{I}}|_R$. For compartment II, we then find by solving the Laplace equation $\mu_{\mathrm{m}}^{\mathrm{II}}(r) = \mu_{\mathrm{m}}^{\mathrm{II}} + \left(\mu_{\mathrm{m}}^{\mathrm{II}}|_R - \mu_{\mathrm{m}}^{\mathrm{II}}\right)R/r$ where $\mu_{\mathrm{m}}^{\mathrm{II}}$ is the chemical potential far from the interface. The flux at the interface can be calculated by $\boldsymbol{j}^{\mathrm{II}}|_R = -\boldsymbol{e}_r m_{\mathrm{m}}\xi^{\mathrm{II}}\partial_r\mu_{\mathrm{m}}^{\mathrm{II}}(r)|_{r=R} = \boldsymbol{e}_r m_{\mathrm{m}}\xi^{\mathrm{II}}\left(\mu_{\mathrm{m}}^{\mathrm{II}}|_R - \mu_{\mathrm{m}}^{\mathrm{II}}\right)/R$. The absence of aggregation at the interface allows that diffusion can equilibrate the chemical potentials, that is $\mu_{\mathrm{m}}^{\mathrm{II}}|_R = \mu_{\mathrm{m}}^{\mathrm{I}}|_R$ which is also equal to the chemical potential deeply inside the phase $\mu_{\mathrm{m}}^{\mathrm{I}}$ because phase I is considered to be homogeneous. The net flux of monomers, $J = J_{\mathrm{r}}^{\mathrm{I}} - J_{\mathrm{r}}^{\mathrm{II}}$ with $J_{\mathrm{r}}^{\mathrm{I}} = 0$, is

$$J = -4\pi R m_{\mathrm{m}}\xi^{\mathrm{II}}\left(\mu_{\mathrm{m}}^{\mathrm{I}} - \mu_{\mathrm{m}}^{\mathrm{II}}\right). \tag{16}$$

Using the chemical potential of the monomers, **Equation (10b)**, we can approximate the chemical potential difference as

$$\mu_{\mathrm{m}}^{\mathrm{I}} - \mu_{\mathrm{m}}^{\mathrm{II}} = k_{\mathrm{B}}T\left[\ln\left(M_{\mathrm{m}}^{\mathrm{I}}\right) - \ln\left(\Gamma M_{\mathrm{m}}^{\mathrm{II}}\right)\right] \tag{17}$$

$$\simeq k_{\mathrm{B}}T\frac{\delta M_{\mathrm{m}}^{\mathrm{I}} - \Gamma\delta M_{\mathrm{m}}^{\mathrm{II}}}{M_{\mathrm{m,eq}}^{\mathrm{I}}}\,, \tag{18}$$

where we expanded $M_{\mathrm{m}}^{\mathrm{I}} = M_{\mathrm{m,eq}}^{\mathrm{I}} + \delta M_{\mathrm{m}}^{\mathrm{I}}$ and $M_{\mathrm{m}}^{\mathrm{II}} = M_{\mathrm{m,eq}}^{\mathrm{II}} + \delta M_{\mathrm{m}}^{\mathrm{II}}$ up to linear order around the equilibrium concentrations $M_{\mathrm{m,eq}}^{\mathrm{I}}$ and $M_{\mathrm{m,eq}}^{\mathrm{II}} = M_{\mathrm{m,eq}}^{\mathrm{I}}/\Gamma$, respectively. Thus net change of monomers due to the exchange of material through the interface reads

$$J \simeq -4\pi R D_{\mathrm{m}}\left(M_{\mathrm{m}}^{\mathrm{I}} - \Gamma M_{\mathrm{m}}^{\mathrm{II}}\right) \tag{19}$$

$$= -k(R)\left(M_{\mathrm{m}}^{\mathrm{I}} - \Gamma M_{\mathrm{m}}^{\mathrm{II}}\right), \tag{20}$$

where the diffusion constant in phase II is $D_{\mathrm{m}} = k_{\mathrm{B}}T m_{\mathrm{m}}\xi^{\mathrm{II}}/M_{\mathrm{m,eq}}^{\mathrm{I}}$ and the rate to relax back to monomer partitioning at equilibrium is $k(R) = 4\pi R D_{\mathrm{m}}$. To ease notation we omitted the "$\delta$" to indicate linear deviations from equilibrium.

## Appendix 3

DOI: https://doi.org/10.7554/eLife.42315.008

# Analytical solution to aggregation kinetics with liquid compartments

In this appendix, we discuss the detail associated with the derivation of analytical solutions to the aggregation kinetics in the presence of liquid compartments, *Equation(3)*, main text.

### Initial layer dynamics

Due to the separation of timescales between monomer equilibration between the two compartments and protein aggregation, the system described by *Equation (3)* (main text) will develop initially through a rapid phase of equilibration, before any aggregation occurs in either compartment. During this initial layer phase, which is the temporal equivalent of a boundary layer (*Astarita and Marucci, 1974*; *Danckwerts, 1951*), the initial values of the monomer concentration in each compartment, $M_{\mathrm{m}}^{\mathrm{I}}(0)$ and $M_{\mathrm{m}}^{\mathrm{II}}(0)$, relax quickly to equilibrium such that the condition $M_{\mathrm{m}}^{\mathrm{I}}(t) = \Gamma M_{\mathrm{m}}^{\mathrm{II}}(t)$ is satisfied before aggregation is initiated. This early equilibration kinetics is described by setting the aggregation terms in *Equation (3)* (main text) to zero, yielding the following equations:

$$\frac{dM_{\mathrm{a}}^{\mathrm{I}}(t)}{\mathrm{dt}} = -k^{\mathrm{I}}\left[M_{\mathrm{m}}^{\mathrm{I}}(t) - \Gamma M_{\mathrm{m}}^{\mathrm{II}}(t)\right],\tag{21a}$$

$$\frac{\mathrm{d}M_{\mathrm{a}}^{\mathrm{II}}(\mathrm{t})}{\mathrm{dt}} = k^{\mathrm{II}}\left[M_{\mathrm{m}}^{\mathrm{I}}(t) - \Gamma M_{\mathrm{m}}^{\mathrm{II}}(t)\right].\tag{21b}$$

The solution to *Equation (21)* is:

$$M_{\mathrm{m}}^{\mathrm{I}} = \Gamma B + \frac{k^{\mathrm{I}}\left[M_{\mathrm{m}}^{\mathrm{I}}(0) - \Gamma M_{\mathrm{m}}^{\mathrm{II}}(0)\right]}{A}e^{-At},\tag{22a}$$

$$M_{\mathrm{m}}^{\mathrm{II}} = B + \frac{k^{\mathrm{II}}\left[M_{\mathrm{m}}^{\mathrm{I}}(0) - \Gamma M_{\mathrm{m}}^{\mathrm{II}}(0)\right]}{A}e^{-At},\tag{22b}$$

where $A = k^{\mathrm{I}} + \Gamma k^{\mathrm{II}}$, and $B = [k^{\mathrm{II}}M_{\mathrm{m}}^{\mathrm{I}}(0) + k^{\mathrm{I}}M_{\mathrm{m}}^{\mathrm{II}}(0)]/A$. Note that the kinetics described by *Equation (22)* 'pushes' the system towards the slow manifold, which is described by $M_{\mathrm{m}}^{\mathrm{I}}(t) = \Gamma M_{\mathrm{m}}^{\mathrm{II}}(t)$. Hence, when $M_{\mathrm{m}}^{\mathrm{I}}(0) = \Gamma M_{\mathrm{m}}^{\mathrm{II}}(0)$, there is no initial phase of 'correction' of the initial conditions. At the end of this initial boundary layer phase, the monomer concentrations in the two compartments are given by:

$$M_{\mathrm{m}}^{\mathrm{I}} = \Gamma B,\tag{23a}$$

$$M_{\mathrm{m}}^{\mathrm{II}} = B.\tag{23b}$$

Since we are not very much interested in this initial phase of redistribution of the initial conditions, in the following we shall assume for simplicity that the initial monomer concentrations in compartments I and II satisfy the relationship $M_{\mathrm{m}}^{\mathrm{I}}(0) = \Gamma M_{\mathrm{m}}^{\mathrm{II}}(0)$. This assumption does not affect the generality of our results. In fact, if this condition was not satisfied initially, then, according to *Equation (22)*, rapid equilibration between the two compartments would correct these initial conditions, eventually leading to a 'corrected' set of initial conditions that lie in the slow manifold.

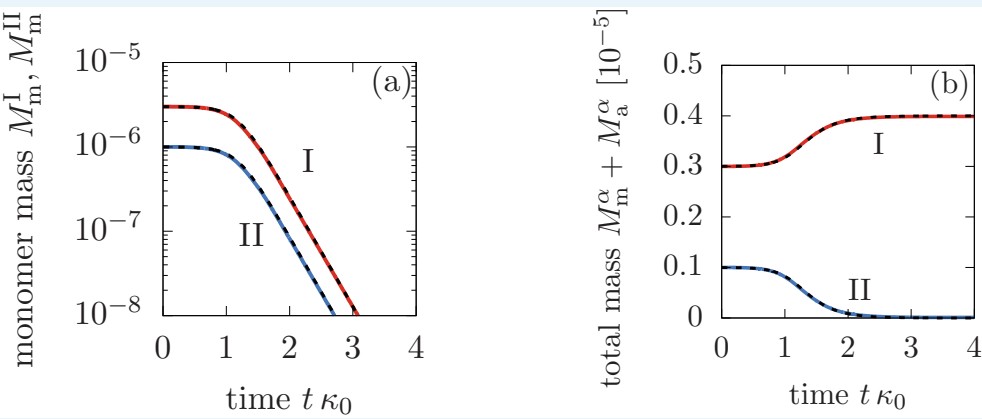

**Appendix 3—figure 1.** Comparison between the analytical solutions *Equations (46), (47), (48)* (dashed lines) and the numerical solution to *Equation (3)* in the main text (solid lines). Panel (**a**) shows the monomer mass concentration in phase I and II, while (**b**) depicts the total monomer mass in form of monomers and aggregate mass in each phase. The parameters are: $k_+ = 10^6$ $M^{-1}s^{-1}$, $k_1 = 10^{-4}$ $M^{-1}s^{-1}$, $k_2 = 10^4$ $M^{-2}s^{-1}$, $M_m^{tot} = 4M$, $n_1 = n_2 = 2$, $\Gamma = 3$, $\xi = 1$ and $k^\alpha/\kappa_0 = 100$ for $\alpha = I, II$.

DOI: https://doi.org/10.7554/eLife.42315.011

## Solving aggregation kinetics in the slow manifold

After an initial, rapid phase of monomer redistribution through the two compartments, the system enters a slower phase of dynamics, where, at leading order, the system stays on the slow manifold $M_m^I = \Gamma M_m^{II}$ at all times. For simplicity, let us assume that the initial concentrations of monomers in the two compartments obey the relationship $M_m^I(0) = \Gamma M_m^{II}(0)$ (otherwise there will be a fast equilibration of the initial conditions such that this relationship is satisfied). To describe the aggregation process in the slow manifold, we write $M_m^I - \Gamma M_m^{II} \simeq 0$ for all times in *Equation (3)* (main text) and find:

$$\frac{dM_m^I(t)}{dt} = -2k_+ M_m^I(t) c_a^I(t) = -\frac{dM_a^I(t)}{dt}, \tag{24a}$$

$$\frac{dM_m^{II}(t)}{dt} = -2k_+ M_m^{II}(t) c_a^{II}(t) = -\frac{dM_a^{II}(t)}{dt}, \tag{24b}$$

$$\frac{dc_a^I(t)}{dt} = k_1 M_m^I(t)^{n_1} + k_2 M_m^I(t)^{n_2} M_a^I(t), \tag{24c}$$

$$\frac{dc_a^{II}(t)}{dt} = k_1 M_m^{II}(t)^{n_1} + k_2 M_m^{II}(t)^{n_2} M_a^{II}(t). \tag{24d}$$

It is useful to introduce the total monomer concentration in the system as:

$$M_m(t) = \frac{V^I M_m^I(t) + V^{II} M_m^{II}(t)}{V}. \tag{25}$$

Note that this concentration may vary in time as aggregates are nucleated and grow, however, the total mass concentration

$$M_m^{tot} = M_m(t) + \frac{V^I M_a^I(t) + V^{II} M_a^{II}(t)}{V} \tag{26}$$

is conserved at all times. Using the condition $M_m^I = \Gamma M_m^{II}$, we can write the following relationships linking the concentrations of monomers in compartments I and II to the total concentration of monomers in the system:

$$M_{\mathrm{m}}^{\mathrm{I}}(t) = \xi \Gamma M_{\mathrm{m}}(t) \,, \tag{27a}$$

$$M_{\mathrm{m}}^{\mathrm{II}}(t) = \xi M_{\mathrm{m}}(t) \,, \tag{27b}$$

where we abbreviated the partitioning degree

$$\xi = \frac{V}{(\Gamma - 1)V^I + V}. \tag{28}$$

Thus, using *Equation (27)*, we can re-write *Equation (24)* as:

$$\frac{\mathrm{dM}_{\mathrm{a}}^{\mathrm{I}}(t)}{\mathrm{dt}} = 2k_+ (\xi \Gamma) M_{\mathrm{m}}(t) c_{\mathrm{a}}^{\mathrm{I}}(t) \,, \tag{29a}$$

$$\frac{\mathrm{dM}_{\mathrm{a}}^{\mathrm{II}}(t)}{\mathrm{dt}} = 2k_+ \xi M_{\mathrm{m}}(t) c_{\mathrm{a}}^{\mathrm{II}}(t) \,, \tag{29b}$$

$$\frac{\mathrm{dc}_{\mathrm{a}}^{\mathrm{I}}(t)}{\mathrm{dt}} = k_1 (\xi \Gamma)^{n_1} M_{\mathrm{m}}(t)^{n_1} + k_2 (\xi \Gamma)^{n_2} M_{\mathrm{m}}(t)^{n_2} M_{\mathrm{a}}^{\mathrm{I}}(t) \,, \tag{29c}$$

$$\frac{\mathrm{dc}_{\mathrm{a}}^{\mathrm{II}}(t)}{\mathrm{dt}} = k_1 \xi^{n_1} M_{\mathrm{m}}(t)^{n_1} + k_2 \xi^{n_2} M_{\mathrm{m}}(t)^{n_2} M_{\mathrm{a}}^{\mathrm{II}}(t) \,. \tag{29d}$$

## Early-time dynamics for aggregate number and mass concentrations in compartments I and II

Before discussing the full time course of aggregation, it is useful to consider the early-time kinetics of the system, which emerges when the monomers in the system have not been depleted significantly (*Michaels et al., 2018*). This limit is obtained by assuming in *Equation (29)* that the total monomer concentration is constant in time, that is $M_{\mathrm{m}}(t) \approx M_{\mathrm{m}}^{\mathrm{tot}}$ (*Michaels et al., 2018*). This assumption transforms the kinetic *Equation (29)* into the following simpler set of linear differential equations:

$$\frac{\mathrm{dM}_{\mathrm{a}}^{\mathrm{I}}(t)}{\mathrm{dt}} = \mu_{\mathrm{I}} c_{\mathrm{a}}^{\mathrm{I}}(t) \,, \tag{30a}$$

$$\frac{\mathrm{dM}_{\mathrm{a}}^{\mathrm{II}}(t)}{\mathrm{dt}} = \mu_{\mathrm{II}} c_{\mathrm{a}}^{\mathrm{II}}(t) \,, \tag{30b}$$

$$\frac{\mathrm{dc}_{\mathrm{a}}^{\mathrm{I}}(t)}{\mathrm{dt}} = \nu_{\mathrm{I}} + \beta_{\mathrm{I}} M_{\mathrm{a}}^{\mathrm{I}}(t) \,, \tag{30c}$$

$$\frac{\mathrm{dc}_{\mathrm{a}}^{\mathrm{II}}(t)}{\mathrm{dt}} = \nu_{\mathrm{II}} + \beta_{\mathrm{II}} M_{\mathrm{a}}^{\mathrm{II}}(t) \,, \tag{30d}$$

where we have introduced the parameters:

$$\mu_{\mathrm{I}} = \mu_0 \xi \Gamma \,, \nu_{\mathrm{I}} = \nu_0 (\xi \Gamma)^{n_1} \,, \beta_{\mathrm{I}} = \beta_0 (\xi \Gamma)^{n_2} \,, \tag{31}$$

and

$$\mu_{\mathrm{II}} = \mu_0 \xi \,, \nu_{\mathrm{II}} = \nu_0 \xi^{n_1} \,, \beta_{\mathrm{II}} = \beta_0 \xi^{n_2} \,, \tag{32}$$

with

$$\nu_0 = k_1 (M_m^{\text{tot}})^{n_1},\tag{33a}$$

$$\beta_0 = k_2 (M_m^{\text{tot}})^{n_2},\tag{33b}$$

$$\mu_0 = 2k_+ M_m^{\text{tot}}.\tag{33c}$$

The solution to **Equation (30)** subject to the condition that no aggregates are present initially reads

$$\frac{M_a^{\text{I}}(t)}{M_m^{\text{I}}(0)} = \frac{\lambda_{\text{I}}^2 [\cosh(\kappa_{\text{I}} t) - 1]}{\kappa_{\text{I}}^2},\tag{34a}$$

$$\frac{M_a^{\text{II}}(t)}{M_m^{\text{II}}(0)} = \frac{\lambda_{\text{II}}^2 [\cosh(\kappa_{\text{II}} t) - 1]}{\kappa_{\text{II}}^2},\tag{34b}$$

for the aggregate mass concentrations in compartments I and II, and

$$c_a^{\text{I}}(t) = \frac{\nu_{\text{I}} \sinh(\kappa_{\text{I}} t)}{\kappa_{\text{I}}}\tag{35a}$$

$$c_a^{\text{II}}(t) = \frac{\nu_{\text{II}} \sinh(\kappa_{\text{II}} t)}{\kappa_{\text{II}}}\tag{35b}$$

for the aggregate number concentrations in compartments I and II. Here, we have introduced the kinetic coefficients

$$\lambda_{\text{I}} = \lambda_0 (\xi \Gamma)^{\frac{n_1}{2}}, \quad \lambda_{\text{II}} = \lambda_0 \xi^{\frac{n_1}{2}},\tag{36}$$

and

$$\kappa_{\text{I}} = \kappa_0 (\xi \Gamma)^{\frac{n_2+1}{2}}, \quad \kappa_{\text{II}} = \kappa_0 \xi^{\frac{n_2+1}{2}},\tag{37}$$

with

$$\lambda_0 = \sqrt{2k_+ k_1 (M_m^{\text{tot}})^{n_1}},\tag{38a}$$

$$\kappa_0 = \sqrt{2k_+ k_2 (M_m^{\text{tot}})^{n_2+1}},\tag{38b}$$

being the effective rates characterizing the proliferation of aggregates due to primary and secondary nucleation, respectively (**Michaels et al., 2018**). According to **Equations (34) and (35)**, the aggregate number and mass concentrations in both compartments grow exponentially with time. The effective growth rates $\kappa_{\text{I}}$ and $\kappa_{\text{II}}$ are different for each compartment and depend on $\Gamma$ and $\xi$. Since $\Gamma \gg 1$, aggregate growth in the early times is much faster in compartment I compared to compartment II. In particular, the ratio of the growth rates in the two compartments is independent of $\xi$ and is given by:

$$\frac{\kappa_{\text{I}}}{\kappa_{\text{II}}} = \Gamma^{\frac{n_2+1}{2}}.\tag{39}$$

Also primary nucleation is enhanced inside compartment I relative to compartment II:

$$\frac{\nu_{\text{I}}}{\nu_{\text{II}}} = \Gamma^{n_1}.\tag{40}$$

## Analytical solution for full time course of monomer concentrations in compartments I and II

We now construct analytical solutions for the monomer and aggregate mass concentrations that are valid for the entire duration of the aggregation reaction. In the previous section, we have seen that for $\Gamma \gg 1$ two timescales, $1/\kappa_I$ and $1/\kappa_{II}$, characterize the early-time aggregation in the two compartments. Since the growth rate in compartment I, $\kappa_I$, is much larger than that in compartment II, $\kappa_{II}$, monomers in compartment I will be consumed by aggregation much faster than those in compartment II. However, the relationship $M_m^I(t) = \Gamma M_m^{II}(t)$ must hold at all times. Thus, to compensate the fast aggregation in compartment I, there will be a flux of monomers from compartment II to compartment I. Eventually, the vast majority of monomers will end up as part of aggregates in compartment I and the parameter $\kappa_I$ will naturally control the depletion of monomers in both compartments. We can make this argument more quantitative by using *Equation (34)* as follows. Monomers in compartment I are consumed over a timescale of the order $1/\kappa_I$. The amount of aggregate mass that will be formed in compartment II during this time period will be of the order

$$M_a^{II} \simeq M_m^{II}(0) \frac{\lambda_{II}^2 [\cosh(\kappa_{II}/\kappa_I) - 1]}{\kappa_{II}^2}. \tag{41}$$

Since $\kappa_{II}/\kappa_I \ll 1$, we can expand the cosh function as a Taylor series, $\cosh x = 1 + x^2/2 + \mathcal{O}(x^5)$. At leading order, we find:

$$M_a^{II} \simeq M_m^{II}(0) \frac{\lambda_{II}^2}{2\kappa_I^2}. \tag{42}$$

Thus, the ratio between the mass of aggregates formed in compartments I and II over a timescale $1/\kappa_I$ is

$$\frac{M_a^I}{M_a^{II}} \simeq \frac{M_a^I(0)}{M_a^{II}(0)} \frac{\lambda_I^2}{\lambda_{II}^2} \simeq \Gamma^{\frac{n_1}{2}+1}. \tag{43}$$

Since $\Gamma \gg 1$, the aggregate mass in compartment I will be much larger than that in compartment II. We can thus neglect at leading order the contribution from $M_a^{II}(t)$ to the conservation of total mass relationship. Doing so, and using *Equation (27)*, we can write the conservation of mass relationship as follows

$$M_a^I(t) = [M_m^I(0) - M_m^I(t)] \frac{\Gamma + 1}{\Gamma}. \tag{44}$$

Using *Equation (44)*, we can reduce the kinetic *Equations (29)* to a system of two coupled different equations:

$$\frac{dM_m^I(t)}{dt} = -2\tilde{k}_+ M_m^I(t) c_a^I(t), \tag{45a}$$

$$\frac{dc_a^I(t)}{dt} = k_1 M_m^I(t)^{n_1} + \tilde{k}_2 M_m^I(t)^{n_2} [M_m^I(0) - M_m^I(t)], \tag{45b}$$

where $\tilde{k}_+ = k_+ \Gamma/(\Gamma + 1)$ and $\tilde{k}_2 = k_2(\Gamma + 1)/\Gamma$. Conveniently, *Equation (45)* are exactly the fundamental kinetic equations describing the dynamics of protein aggregation in a pure system, that is without compartment, but with effective rate parameters that depend on the degree of phase separation (*Michaels et al., 2018*; *Michaels et al., 2016*). Thus, we can adapting the results in *Michaels et al. (2016)* to *Equation (45)*, we find the following solution for the time varying monomer concentration in compartment I:

$$\frac{M_{\mathrm{m}}^{\mathrm{I}}(t)}{M_{\mathrm{m}}^{\mathrm{I}}(0)} = \left[1 + \frac{\lambda_{\mathrm{I}}^2}{2\kappa_{\mathrm{I}}^2\theta}\left(\frac{\Gamma}{\Gamma+1}\right)e^{\kappa_{\mathrm{I}}t}\right]^{-\theta},$$

(46)

where $\theta = \sqrt{2/[n_2(n_2+1)]}$. Using **Equation (44)**, we then obtain an expression for the aggregate mass concentration:

$$\frac{M_{\mathrm{a}}^{\mathrm{I}}(t)}{M_{\mathrm{m}}^{\mathrm{I}}(0)} = \frac{\Gamma+1}{\Gamma}\left(1 - \left[1 + \frac{\lambda_{\mathrm{I}}^2}{2\kappa_{\mathrm{I}}^2\theta}\left(\frac{\Gamma}{\Gamma+1}\right)e^{\kappa_{\mathrm{I}}t}\right]^{-\theta}\right).$$

(47)

Finally, the time course of the monomer concentration in compartment II is obtained using the relationship $M_{\mathrm{m}}^{\mathrm{I}}(t) = \Gamma M_{\mathrm{m}}^{\mathrm{II}}(t)$. This yields:

$$\frac{M_{\mathrm{m}}^{\mathrm{II}}(t)}{M_{\mathrm{m}}^{\mathrm{II}}(0)} = \left[1 + \frac{\lambda_{\mathrm{I}}^2}{2\kappa_{\mathrm{I}}^2\theta}\left(\frac{\Gamma}{\Gamma+1}\right)e^{\kappa_{\mathrm{I}}t}\right]^{-\theta}.$$

(48)

The accuracy of our analytical solutions **Equations (46), (47) and (48)** against numerical integration of **Equation (3)** (main text) is shown in **Figure 1**.

## Scaling relationships for the aggregate number concentrations in compartments I and II

From the knowledge of the time varying monomer concentration, **Equation (46)**, we can obtain an expression for the aggregate number concentration in compartment I using **Equation (45a)** by simple differentiation of **Equation (46)**, $c_{\mathrm{a}}^{\mathrm{I}}(t) = -1/[2\tilde{k}_+ M_{\mathrm{m}}^{\mathrm{I}}(t)]\mathrm{d}M_{\mathrm{m}}^{\mathrm{I}}(t)/\mathrm{d}t$. This yields the following expression:

$$\frac{c_{\mathrm{a}}^{\mathrm{I}}(t)}{c_{\mathrm{a}}^{\mathrm{I}}(\infty)} = \left[1 + \frac{2\kappa_{\mathrm{I}}^2\theta}{\lambda_{\mathrm{I}}^2}\left(\frac{\Gamma+1}{\Gamma}\right)e^{-\kappa_{\mathrm{I}}t}\right]^{-1},$$

(49)

where

$$c_{\mathrm{a}}^{\mathrm{I}}(\infty) = \frac{\kappa_{\mathrm{I}}\theta}{2k_+}\left(\frac{\Gamma+1}{\Gamma}\right)$$

(50)

is the number concentration of aggregates at steady state. It is interesting to extract from **Equation (50)** the key dependence of $c_{\mathrm{a}}^{\mathrm{I}}(\infty)$ on the parameters $\xi$ and $\Gamma$:

$$c_{\mathrm{a}}^{\mathrm{I}}(\infty) = \frac{\kappa_0\theta}{2k_+}\xi^{\frac{n_2+1}{2}}\Gamma^{\frac{n_2-1}{2}}(\Gamma+1).$$

(51)

Note that the prefactor defines the homogeneous concentration in the absence of compartments,

$$c_{\mathrm{a}}^{\mathrm{tot}}\big|_{\phi^{\mathrm{I}}=\phi^{\mathrm{II}}} = \frac{\kappa_0\theta}{2k_+}.$$

(52)

Thus, using $\Gamma \gg 1$ we find the following scaling relationship for the steady-state number concentration of aggregates in compartment I:

$$c_{\mathrm{a}}^{\mathrm{I}}(\infty) \simeq c_{\mathrm{a}}^{\mathrm{tot}}\big|_{\phi^{\mathrm{I}}=\phi^{\mathrm{II}}}(\xi\Gamma)^{\frac{n_2+1}{2}}.$$

(53)

The scaling relationship **Equation (53)** can be rationalized as follows. The terminal concentration of aggregates in compartment I is given as $c_{\mathrm{a}}^{\mathrm{I}}(\infty) \simeq M_{\mathrm{a}}^{\mathrm{I}}(\infty)/L^{\mathrm{I}}(\infty)$, where $L^{\mathrm{I}}(\infty)$ is the average size of aggregates in compartment I. $L^{\mathrm{I}}(\infty)$ is determined by the ratio between aggregate growth and nucleation as $\simeq \sqrt{\mu^{\mathrm{I}}/\beta^{\mathrm{I}}}$ (**Michaels and Knowles, 2014**; **Michaels et al., 2015**); indeed, increasing the rate of growth over the rate of nucleation leads to more monomers being consumed in aggregate elongation, hence to longer aggregates. Since

$M_{\mathrm{a}}^{\mathrm{I}}(\infty)/M_{\mathrm{m}}^{\mathrm{tot}} \simeq \xi\Gamma$ and using **Equation (31)**, we find $L^{\mathrm{I}}(\infty) \simeq (\xi\Gamma)^{\frac{1-n_2}{2}}$ and, hence, $c_{\mathrm{a}}^{\mathrm{I}}(\infty) \simeq (\xi\Gamma)/(\xi\Gamma)^{\frac{1-n_2}{2}} \simeq (\xi\Gamma)^{\frac{n_2+1}{2}}$.

A similar scaling relationship to **Equation (53)** can be derived also for the steady-state number concentration of aggregates in compartment II as follows. We recall that the early-time dynamics of aggregation in compartment II is characterized by a timescale $1/\kappa_{\mathrm{II}}$, which is much slower than the timescale of aggregation in compartment I, $1/\kappa_{\mathrm{I}}$. Thus, $c_{\mathrm{a}}^{II}$ can be considered to be still in the exponential growing phase even when the aggregate concentration in compartment I is equilibrating. Eventually, the assembly in compartment II is arrested abruptly as soon as aggregation in compartment I is fully saturated, since no monomer is left in either compartment. Since the timescale for saturation of aggregation in compartment I is proportional to $1/\kappa_{\mathrm{I}}$ (see **Equation 46**), we can estimate the concentration of aggregates in compartment II at the end of the reaction as:

$$c_{\mathrm{a}}^{\mathrm{II}}(\infty) \simeq \frac{\nu_{\mathrm{II}} \sinh(\kappa_{\mathrm{II}}/\kappa_{\mathrm{I}})}{\kappa_{\mathrm{II}}} = \frac{\nu_{\mathrm{II}}}{\kappa_{\mathrm{II}}} \sinh\left(\Gamma^{-\frac{n_2+1}{2}}\right), \tag{54}$$

where in the last step we used **Equation (39)**. Since we are interested in the limit of relatively large monomer partitioning $\Gamma \gg 1$, the argument of the $\sinh$ function is much smaller than unity. Hence, we can simplify **Equation (54)** by using a Taylor expansion of the $\sinh$ function to first order, $\sinh x = x + \mathcal{O}(x^3)$, yielding the following scaling relationship after extracting the $\xi$ dependence of $\nu_{\mathrm{II}}$ and $\kappa_{\mathrm{II}}$:

$$c_{\mathrm{a}}^{\mathrm{II}}(\infty) \simeq c_{\mathrm{a}}^{\mathrm{tot}}|_{\phi^{\mathrm{I}}=\phi^{\mathrm{II}}} w\, \xi^{n_1-\frac{n_2+1}{2}} \Gamma^{-\frac{n_2+1}{2}}, \tag{55}$$

where $w = k_1/(k_2\theta)(M_{\mathrm{m}}^{\mathrm{tot}})^{n_1-n_2-1}$. Combining **Equation (53)** with **Equation (55)**, we obtain one of the key results of our paper, namely the scaling behavior of aggregate partitioning between compartments I and II with $\Gamma$:

$$\frac{c_{\mathrm{a}}^{\mathrm{I}}(\infty)}{c_{\mathrm{a}}^{\mathrm{II}}(\infty)} \propto \xi^{n_2+1-n_1} \Gamma^{n_2+1}, \tag{56}$$

where the impact of compartment volume on the relative degree of monomer characterized by $\xi(\bar{\phi})$ is given in **Equation (28)**.

## Effect of different rates inside and outside the compartment

So far, we have assumed constant growth rates inside and outside of the compartment; in general, this assumption will not hold for all phase separating systems combined with aggregation. We can extend our framework to take into account the effect of different rates for the condensed ($\alpha = \mathrm{I}$) and dilute ($\alpha = \mathrm{II}$) phases:

$$\frac{\mathrm{d}c_{\mathrm{a}}^{\alpha}(t)}{\mathrm{d}t} = k_1^{\alpha} M_{\mathrm{m}}^{\alpha}(t)^{n_1} + k_2^{\alpha} M_{\mathrm{m}}^{\alpha}(t)^{n_2} M_{\mathrm{a}}^{\alpha}(t), \tag{57a}$$

$$\frac{\mathrm{d}M_{\mathrm{a}}^{\alpha}(t)}{\mathrm{d}t} = 2k_+^{\alpha} M_{\mathrm{m}}^{\alpha}(t)\, c_{\mathrm{a}}^{\alpha}(t), \tag{57b}$$

$$\frac{\mathrm{d}M_{\mathrm{m}}^{\alpha}(t)}{\mathrm{d}t} = -2k_+^{\alpha} M_{\mathrm{m}}^{\alpha}(t)\, c_{\mathrm{a}}^{\alpha}(t) + \frac{J^{\alpha}}{V^{\alpha}}, \tag{57c}$$

and can then follow the same approach as for the case of homogeneous rates to determine how the presence of different rates affects the various scaling relationships discussed above. For instance, **Equations (39) and (40)** become:

$$\frac{\kappa_{\mathrm{I}}}{\kappa_{\mathrm{II}}} = \sqrt{\frac{k_+^{\mathrm{I}} k_2^{\mathrm{I}}}{k_+^{\mathrm{II}} k_2^{\mathrm{II}}}} \, \Gamma^{\frac{n_2+1}{2}} \tag{58a}$$

$$\frac{\nu_{\mathrm{I}}}{\nu_{\mathrm{II}}} = \frac{k_1^{\mathrm{I}}}{k_1^{\mathrm{II}}} \, \Gamma^{n_1}. \tag{58b}$$

Moreover, *Equations (53) and (55)* become:

$$c_{\mathrm{a}}^{\mathrm{I}}(\infty) \simeq \sqrt{\frac{k_2^{\mathrm{I}}}{2k_+^{\mathrm{I}}}} \left(M_{\mathrm{m}}^{\mathrm{tot}}\right)^{\frac{n_2+1}{2}} \theta \left(\xi\Gamma\right)^{\frac{n_2+1}{2}} \tag{59a}$$

$$c_{\mathrm{a}}^{\mathrm{II}}(\infty) \simeq \frac{k_1^{\mathrm{II}}}{\sqrt{2k_+^{\mathrm{II}} k_2^{\mathrm{II}}}} \left(M_{\mathrm{m}}^{\mathrm{tot}}\right)^{n_1 - \frac{n_2+1}{2}} \xi^{n_1 - \frac{n_2+1}{2}} \Gamma^{-\frac{n_2+1}{2}}, \tag{59b}$$

such that

$$\frac{c_{\mathrm{a}}^{\mathrm{I}}(\infty)}{c_{\mathrm{a}}^{\mathrm{II}}(\infty)} \simeq \sqrt{\frac{k_+^{\mathrm{II}}}{k_+^{\mathrm{I}}}} \, \frac{\sqrt{k_2^{\mathrm{I}} k_2^{\mathrm{II}}}}{k_1^{\mathrm{II}}} \left(M_{\mathrm{m}}^{\mathrm{tot}}\right)^{n_2+1-n_1} \theta \, \xi^{n_2+1-n_1} \Gamma^{n_2+1}. \tag{59c}$$

Note that there is an asymmetry in the dependence of $c_{\mathrm{a}}^{\mathrm{I}}(\infty)$ and $c_{\mathrm{a}}^{\mathrm{II}}(\infty)$ on the rate constants of primary and secondary nucleation; it originates from the fact that, due to separation of timescales and due to $\Gamma \ll 1$, aggregation in compartment II is arrested abruptly as soon as aggregation in I reaches saturation (see *Equation 54*). This effect makes $c_{\mathrm{a}}^{\mathrm{II}}(\infty)$ depend on the rate of primary nucleation. Importantly, from the expressions *Equation (59)*, we see that different growth rates of the compartments enter as multiplicative pre-factors. Hence, even though having different rates for the condensed and dilute phases can certainly affect aggregate partitioning (enhanced partitioning occurs for larger growth rates in I, and vice versa), the effect on the rate constants must be sufficiently large (at least one oder of magnitude) to dominate the effect of monomer partitioning $\Gamma$. This suggest that the mechanism of aggregate partitioning proposed in this study is likely to be robust if aggregation rates differ among the compartments.

