## [Decision Letter]

Thank you for submitting your article "Spatial control of irreversible protein aggregation" for consideration by *eLife*. Your article has been reviewed by three peer reviewers, and the evaluation has been overseen by Frank Jülicher as Reviewing Editor and Arup Chakraborty as the Senior Editor. The following individual involved in the review of your submission has agreed to reveal his identity: Alexander Grosberg (Reviewer #3).

The reviewers have discussed the reviews with one another and the Reviewing Editor has drafted this decision to help you prepare a revised submission.

Summary:

The work on "Spatial control of irreversible protein aggregation" presents novel ideas and a theoretical approach to show that protein aggregates could predominantly form in liquid cellular compartments and strongly enrich there. Such compartments could therefore play a role to reduce protein aggregation outside such compartments and to control where irreversible aggregation is permitted to occur. The approach used is simple and elegant. The paper has a number of insights gleaned from both Flory-Huggins and kinetic theories of phase separation and aggregation kinetics, respectively. Presented equations are formulated to provide insight into the rich interplay between monomer partitioning, condensate volume fraction, and aggregation mechanism on the resulting aggregation location and rates under a quantitative framework. The main results are: 1) When monomers that tend to aggregate partition preferentially in a liquid droplet this results in a positive feedback driving aggregation inside droplets, as it causes a flux of monomers from the dilute phase to the droplet. 2) A phase diagram reveals a switch between regions where large or small compartments show maximal aggregate enrichment, respectively. This switch depends on the orders of the primary and secondary aggregate nucleation reactions. This is a strong and interesting paper which presents important theoretical ideas that are relevant for cell biology. The work presented is in principle suitable for *eLife*. However the authors should carefully consider several specific points raised by the reviewers and in a revised manuscript they should formulate testable predictions more explicitly and improve the links to potential experimental tests.

Essential revisions:

1) The main entrance point to the model is Equation 1 (same as 14). It operates with phase segregation between A-rich and A-poor liquid phases in which monomers and aggregates are also dissolved. This idea is not explained well, because terms (monomer, aggregate, A, B, etc.) are used without definitions. More importantly, what are A's and B's? Presumably A's and B's are also proteins of some sort. It is unclear why it is fair to talk about equilibrium and kinetics of proteins called monomers while assuming other proteins (in fact, perhaps many protein species!) to passively maintain equilibrium volume fractions *ϕ^I^* and *ϕ^II^*. One would actually expect that even a small amount of a particular protein called "monomer" could have a significant effect on the balancing of coexisting densities given the multitude of components hidden inside notations A and B.

2) In a regular theory of liquid nuclei (like, e.g., Lifshitz and Slyozov, 1961) an important part of the setup is dependence of equilibrium volume fractions on the nucleus size (the Laplace pressure effect); it is unclear why present theory neglects this altogether. To some extent authors may make an argument that concentration of "monomers" is small and does not affect the droplet size; but as irreversible aggregation continues, aggregates accumulate, and they seem to be bound to start affecting the droplet size eventually.

3) It is also not clear why in the kinetics part it is possible to neglect the fact that one of the phases might be dramatically more viscous than the other. The authors themselves cite works on polymer reputation and even possible solidification, but do not seem to consider the viscosity increase in the subsequent considerations, such as Equation 3. Again, this is possibly justifiable at the initial stage, but perhaps not so easily on a later time. I think that the authors have in mind some self-consistent system of conditions where all these questions may be answered, but they did not present these conditions in a clear way. These conditions and the terminology should be clarified.

4) In the subsection “Mathematical model for liquid compartments controlling protein aggregation”, the authors present an interesting approach to understanding the partition coefficient that is quite useful. However, the authors comment that the partitioning is solely governed by the relative interaction strength *Δχ* between the monomers with the A and the B components. This statement seems overstated, since other issues such as the relative density of the components and the potential of non-*χ* type interactions (e. g. direct binding between A and B) may not captured.

5) The text around Equations 3B-3D needs to be elaborated. *n*_1_ and *n*_2_ need to be defined and the description of the processes being considered requires more detail.

6) It is worth noting that the type of analyses carried out in this paper, including the analogy with Boundary Layer theory, have been considered extensively in the past in the context of gas-liquid reactions that are used extensively in the chemical industry for separation processes. The work of G. Astarita and P.V. Danckwerts are just two examples. Some reference to these studies may be appropriate.

7) It is important to note whether the biological systems placed on the "phase diagram" in Figure 4 are known to correspond to the big or small compartment limits. If not, how might the prediction be tested.

8) More generally the authors should formulate testable predictions more explicitly and improve the links to potential experimental tests.

9) The model assumes that the condensed phase and dilute phase have the same kinetics of growth for aggregation. This is likely inaccurate and thus should be addressed. For example, others have noticed (e.g. Wei et al., 2017), a significant size originated non-ideality in partitioning occurs in many protein-rich phases in vivo and in vitro. This non-ideality would weaken the kinetics of growth for aggregates and potentially limit their size. It is unclear how much such non-ideal contributions would perturb the conclusions thus requiring some discussion.

10) The authors focus on cytoplasm vs. specific biomolecule-dense liquid phases. The authors should not limit themselves to the cytoplasm, since these principles would apply to aggregation in the nucleoplasm as well, and are thus worth at least commenting on.

11) The authors discuss the concept of primary vs. secondary nucleation, and indeed their analysis focuses on key differences in these two modes. But I think they should be introduced, at least in a few sentences, in a qualitative way, prior to diving into the mathematical details.

12) The authors focus on droplet size as a key factor, but the analysis seems to be exclusive with the simplest mapping from volume fraction to droplet size, by taking the condensed phase as all having coalesced into a single droplet. The authors should at least comment on the expected effects of how a distribution of droplet sizes, such as those commonly reported in the intracellular phase separation literature, would manifest in this context.

[Editors' note: further revisions were requested prior to acceptance, as described below.]

Thank you for resubmitting your work entitled "Spatial control of irreversible protein aggregation" for further consideration at *eLife*. Your revised article has been favorably evaluated by Arup Chakraborty (Senior Editor) and a Reviewing Editor.

The manuscript has been improved but there is one remaining issue that needs to be addressed before acceptance, as outlined below:

You have addressed most points raised and have improved the manuscript. However, an important criticism of the reviewers was formulated:

"8) More generally the authors should formulate testable predictions more explicitly and improve the links to potential experimental tests."

The reviewers felt that it was rather unclear how the ideas of the paper could be tested in experiments and whether there are clear predictions from the work for future experiments. In the revision of the manuscript this point was addressed somewhat superficially. This is not easy to do, but we would be grateful if you might try again to address this point more carefully. We anticipate that the paper will be accepted after that.

---

## [Author Response]

Essential revisions:1) The main entrance point to the model is Equation 1 (same as 14). It operates with phase segregation between A-rich and A-poor liquid phases in which monomers and aggregates are also dissolved. This idea is not explained well, because terms (monomer, aggregate, A, B, etc.) are used without definitions. More importantly, what are A's and B's? Presumably A's and B's are also proteins of some sort. It is unclear why it is fair to talk about equilibrium and kinetics of proteins called monomers while assuming other proteins (in fact, perhaps many protein species!) to passively maintain equilibrium volume fractions ϕ^I^ and ϕ^II^. One would actually expect that even a small amount of a particular protein called "monomer" could have a significant effect on the balancing of coexisting densities given the multitude of components hidden inside notations A and B.

The reviewers suggest improving the discussion and presentation of the paragraph about the three component phase separation model and why a consideration at phase separation equilibrium of the A-B mixture is valid. We have now revised the corresponding paragraphs and explicitly mention what species A and B could represent (note, in any case, that an exact definition is unnecessary because only physical parameter enter the model). Moreover, we now briefly explain why the binary phase separation between A and B is well equilibrated while the partitioning of monomer deviates weakly from the partitioning equilibrium. Such assumptions are well satisfied because of two reasons: 1) Monomer species are very dilute and thus cannot affect A-B phase equilibrium. 2) Moreover, aggregation kinetic is slow relative to the partitioning of monomers via diffusion. Thus, at each time point during the aggregation kinetics, monomer partitioning is very close to the partitioning equilibrium. For better overview we have now combined all essential assumptions of our model in the first section of the model description.

2) In a regular theory of liquid nuclei (like, e.g., Lifshitz and Slyozov, 1961) an important part of the setup is dependence of equilibrium volume fractions on the nucleus size (the Laplace pressure effect); it is unclear why present theory neglects this altogether. To some extent authors may make an argument that concentration of "monomers" is small and does not affect the droplet size; but as irreversible aggregation continues, aggregates accumulate, and they seem to be bound to start affecting the droplet size eventually.

We agree with the reviewer that we have neglected the effect of the compartment radius R on the equilibrium concentrations. For a binary A-B mixture, this effect can be captured by the Gibbs-Thomson relationship stating that the relative increase on both equilibrium concentrations inside and outside increases, cαlγα/R, where lγα denotes the capillary length inside and outside. However, the capillary length is typically in the order of a few molecular sizes. Thus, as long as the compartment size significantly exceeds the molecular size (which is the typical case inside cells for example), the actual equilibrium concentration values would differ only very weakly relative to the equilibrium concentrations for large R. Of course, this little concentration difference is key for the coarsening kinetics (Ostwald ripening) where the small fraction, lγα/R, competes with another very small value, namely the supersaturation. However, in our work we only consider one compartment and not the coarsening of many compartments. In the presence of monomers the arguments above apply as well because the considered monomer concentrations are so dilute that we can neglect their impact on the equilibrium concentrations and also on the surface tension (in the case they would act as surfactants). We have added a comment to the main text and a paragraph to the Appendix discussing when Laplace pressure effects can be neglected.

Please note that we have estimated typical values for homogeneous volume fraction of aggregating monomers to be 10^-9^ – 10^-8^. Hence, even for large partitioning factors, such as e.g. Γ=100, the local concentrations of monomers remain dilute; this allows us to safely neglect the contributions of monomers to droplet volume. Based on the reviewer’s comments, we have now commented on the role of Laplace pressure effects in the appendix (see text after Equation 13) and the impact of monomers on droplet in the main text (see end of paragraph “Model for liquid compartments controlling protein aggregation”).

3) It is also not clear why in the kinetics part it is possible to neglect the fact that one of the phases might be dramatically more viscous than the other. The authors themselves cite works on polymer reputation and even possible solidification, but do not seem to consider the viscosity increase in the subsequent considerations, such as Equation 3. Again, this is possibly justifiable at the initial stage, but perhaps not so easily on a later time. I think that the authors have in mind some self-consistent system of conditions where all these questions may be answered, but they did not present these conditions in a clear way. These conditions and the terminology should be clarified.

We agree that very complex rheological properties could emerge in the later stage of the aggregation kinetics. It is correct that our assumption of a constant monomer diffusion coefficient, while aggregates hardly diffuse, may no more be quantitively correct. One may expect that the monomers could pick up a thickening behaviour due to the interactions with the aggregates on long time scales that would lower their diffusivity as a function of time. We have not included these effects in our model. However, we think that they are worth studying for specific cellular and aggregating systems because rheological properties, in particular, time dependent ones, vary from system to system. Please note that in our manuscript, we present a study of a mechanism which is expected to be qualitatively robust against different rheological properties – of course, quantitative changes of the degree of aggregate partitioning may occur. Based on the reviewer’s comment above we have decided to add a comment to the main text and the outlook.

4) In the subsection “Mathematical model for liquid compartments controlling protein aggregation”, the authors present an interesting approach to understanding the partition coefficient that is quite useful. However, the authors comment that the partitioning is solely governed by the relative interaction strength Δχ between the monomers with the A and the B components. This statement seems overstated, since other issues such as the relative density of the components and the potential of non-χ type interactions (e. g. direct binding between A and B) may not captured.

We thank the reviewers for this comment. Our statement only holds for dilute monomer volume fraction (typical case for the considered aggregating systems discussed in our manuscript) and in the absence of binding processes. We have corrected this statement in our manuscript.

5) The text around Equations 3B-3D needs to be elaborated. n_1_ and n_2_ need to be defined and the description of the processes being considered requires more detail.

The reviewers suggest elaborating on the presentation about the primary and secondary reaction orders. Accordingly, we have revised the text around Equations (3B-3D). In particular, we now give a physical interpretation of the reaction orders *n*_1_ and *n*_2_.

6) It is worth noting that the type of analyses carried out in this paper, including the analogy with Boundary Layer theory, have been considered extensively in the past in the context of gas-liquid reactions that are used extensively in the chemical industry for separation processes. The work of G. Astarita and P.V. Danckwerts are just two examples. Some reference to these studies may be appropriate.

We thank the reviewers for suggestions of references on the boundary layer theory. We have included the proposed references to the discussion of boundary-layer dynamics (see text before Equation 21 in the Appendix).

7) It is important to note whether the biological systems placed on the "phase diagram" in Figure 4 are known to correspond to the big or small compartment limits. If not, how might the prediction be tested.

To the best of our knowledge, no experiments are currently available that would allow us to judge which system corresponds to the small or large compartment limit. However, we appreciate the suggestion to think more of how to test the “phase diagram”. We thus revised the paragraph in the conclusion section, where we have already suggested how our prediction could be tested. Now we also include a short discussion related to Figure 4.

8) More generally the authors should formulate testable predictions more explicitly and improve the links to potential experimental tests.

We have revised the paragraphs in the conclusion section, which now present explicit experimental tests. Moreover, we tried to stress the link to experimental systems by structuring the conclusion section more clearly using subheadings.

9) The model assumes that the condensed phase and dilute phase have the same kinetics of growth for aggregation. This is likely inaccurate and thus should be addressed. For example, others have noticed (e.g. Wei et al., 2017), a significant size originated non-ideality in partitioning occurs in many protein-rich phases in vivo and in vitro. This non-ideality would weaken the kinetics of growth for aggregates and potentially limit their size. It is unclear how much such non-ideal contributions would perturb the conclusions thus requiring some discussion.

It is correct that the assumption of equal kinetic parameters inside and outside will not hold for all phase separating systems combined with aggregation. In our manuscript we wanted to make the point that a strong partitioning of aggregates already occurs for equal kinetic parameters inside and outside. To test how deviations from our ideal assumption affect our conclusions we have derived an equation for the partitioning of aggregates in the case of different growth rates inside and outside. We have now added a corresponding discussion to the main text and show the extended equations in the appendix. We also cite the reference mentioned by the reviewers and discuss Brangwynne et al.’s finding of a semi-dilute mesh of proteins forming droplets that can affect physical properties inside the compartment.

10) The authors focus on cytoplasm vs. specific biomolecule-dense liquid phases. The authors should not limit themselves to the cytoplasm, since these principles would apply to aggregation in the nucleoplasm as well, and are thus worth at least commenting on.

Great point. We now also mention the nucleoplasm (see e.g. Abstract).

11) The authors discuss the concept of primary vs. secondary nucleation, and indeed their analysis focuses on key differences in these two modes. But I think they should be introduced, at least in a few sentences, in a qualitative way, prior to diving into the mathematical details.

Following the reviewers’ suggestion, we have included (before Equation 3) a detailed discussion of primary and secondary nucleation in protein aggregation (see also reply to point 5).

12) The authors focus on droplet size as a key factor, but the analysis seems to be exclusive with the simplest mapping from volume fraction to droplet size, by taking the condensed phase as all having coalesced into a single droplet. The authors should at least comment on the expected effects of how a distribution of droplet sizes, such as those commonly reported in the intracellular phase separation literature, would manifest in this context.

This is a fascinating comment. We would expect that either the smallest or the largest droplets in the system would initially enrich more of the aggregates (depending on the reaction orders of primary vs. secondary nucleation). During the aggregation kinetics there would be a competition about the monomers between the droplets of different size. Droplets of different size thus end up with different aggregate fraction. We now mention these thoughts in the conclusion of our manuscript.

[Editors' note: further revisions were requested prior to acceptance, as described below.]You have addressed most points raised and have improved the manuscript. However, an important criticism of the reviewers was formulated:"8) More generally the authors should formulate testable predictions more explicitly and improve the links to potential experimental tests."The reviewers felt that it was rather unclear how the ideas of the paper could be tested in experiments and whether there are clear predictions from the work for future experiments. In the revision of the manuscript this point was addressed somewhat superficially. This is not easy to do, but we would be grateful if you might try again to address this point more carefully. We anticipate that the paper will be accepted after that.

We were delighted by the positive response of the reviewers to our work. We have included suggestions into our manuscript in the context of experimentally testable predictions. We have also added a new short section on a calculation of the amount of aggregates in the compartments – summarized in a new Figure 5, which we think is a complementary perspective on the rest of the paper.